# Systematically developing a registry of splice-site creating variants utilizing massive publicly available transcriptome sequence data

Naoko Iida[1,3], Ai Okada[1,3], Yoshihisa Kobayashi [2,3], Kenichi Chiba[1], Yasushi Yatabe [2] & Yuichi Shiraishi [1] ✉

Genomic variants causing abnormal splicing play important roles in genetic disorders and cancer development. Among them, variants that cause the formation of novel splice-sites (splice-site creating variants, SSCVs) are particularly difficult to identify and often overlooked in genomic studies. Additionally, these SSCVs are frequently considered promising candidates for treatment with splice-switching antisense oligonucleotides (ASOs). To leverage massive transcriptome sequence data such as those available from the Sequence Read Archive, we develop a novel framework to screen for SSCVs solely using transcriptome data. We apply it to 322,072 publicly available transcriptomes and identify 30,130 SSCVs. Among them, 5121 SSCVs affect disease-causing variants. By utilizing this extensive collection of SSCVs, we reveal the characteristics of Alu exonization via SSCVs, especially the hotspots of SSCVs within Alu sequences and their evolutionary relationships. We discover novel gain-of-function SSCVs in the deep intronic region of the *NOTCH1* gene and demonstrate that their activation can be suppressed using splice-switching ASOs. Collectively, we provide a systematic approach for automatically acquiring a registry of SSCVs, which facilitates the elucidation of novel biological mechanisms underlying splicing and serves as a valuable resource for drug discovery. The catalogs of SSCVs identified in this study are accessible on the SSCV DB (https://sscvdb.io).

Genomic variants causing abnormal splicing are among the most important causes of genetic disease and cancer[1,2]. However, assessing the impact of each variant on splicing is by no means a trivial task. To detect and catalog splicing-associated variants, one approach is to identify significant relationships between genomic variants and their associated splicing changes through the analysis of matched genome and transcriptome sequence data[3–8]. Nonetheless, cohorts with both genome and transcriptome data are not common. On the other hand,

the Sequence Read Archive (SRA) provides a massive repository of publicly available transcriptome sequence data[9]. There are increasing efforts to process SRA transcriptome data in a uniform manner, generating resources such as gene and transcript expression[10–12]. Recently, to utilize the SRA transcriptome in the context of detecting splicing-associated variants, we developed a method capable of identifying genomic variants that cause a specific type of abnormal splicing, intron retention, using only transcriptome sequence data. This method was

[1]Division of Genome Analysis Platform Development, National Cancer Center Research Institute, Tokyo, Japan. [2]Division of Molecular Pathology, National Cancer Center Research Institute, Tokyo, Japan. [3]These authors contributed equally: Naoko Iida, Ai Okada, Yoshihisa Kobayashi. ✉e-mail: yuishira@ncc.go.jp

**Fig. 1 | Overview and evaluation of the algorithm for SSCV detection. a** Splicing consequences via SSCVs. **b** Example of alignment status around SSCVs by the depiction of Integrative Genomics Viewer for the SSCVs in *SDHA* (ENST00000264932.11: c.1751 C > T) (left) and in *PRKAR1A* (ENST00000589228.6: c.892-129 C > G) (right). We can observe mismatch bases corresponding to the SSCVs. **c** Overview of juncmut procedures. **d** Overview of the evaluation of juncmut using the 1000 Genomes Project and GTEx genome and transcriptome sequence data. **e** Comparative evaluation of juncmut against SpliceAI. We analyzed variants from the 1000 Genomes Project whole-genome sequencing data with SpliceAI scores exceeding the thresholds (0.2, 0.5, and 0.8), as well as SSCVs identified by juncmut. We selected variants observed in at least one sample within the GTEx cohorts and calculated the combined *p*-value (measuring the difference in abnormal splicing ratios between samples with and without the variant across various tissues; see "Methods" section for details). Each *p*-value corresponding to these variants is plotted accordingly. The numbers of remaining SNVs with SpliceAI scores exceeding 0.2, 0.5, and 0.8 were 7923, 1498, and 336, respectively; additionally, 22 SNVs identified by juncmut were plotted. The boxplot summarizes the combined *p*-values. The ends of the boxes represent the lower and upper quartiles, the center line indicates the median, and the whiskers show the maximum and minimum values within 1.5 × IQR from the edges of the box, respectively. **f** An example of the relative abnormal splicing ratios in the presence (orange) and absence (gray) of SSCVs across various tissues, as measured using GTEx transcriptome data. See also Supplementary fig. 2. **e, f** Source data are provided as a Source Data file.

applied to over 230,988 transcriptome sequence data reposited in the SRA, leading to the cataloging of over 27,049 intron retention associated variants[13].

In this study, we aimed to focus on another highly crucial class of splicing-associated variants, splice-site creating variants (SSCVs), in which abnormal splicing is caused by newly created splice donor/acceptor motifs resulting from the variants. Examples of SSCVs include apparently silent variants in exonic regions that, in fact, create an abrupt splice-site, leading to partial exon loss (left panel of Fig. 1a). Additionally, some variants in deep intronic regions can form novel

splice-sites, resulting in the inclusion of cryptic exons (right panel of Fig. 1a). Compared with splicing-associated variants that disrupt normally used splice donor/acceptor sites (which are naturally concentrated near the exon-intron boundaries), it is more challenging to extract SSCVs since regions in which SSCVs can occur span the entire genetic region. Therefore, the precise identification of SSCVs opens the door to uncovering previously undetected genomic mutations responsible for diseases[14,15]. Furthermore, these SSCVs, especially those located in deep intronic regions, are often regarded as highly suitable targets for splice-switching antisense oligonucleotides (ASOs)[16,17], presenting valuable treatment opportunities for patients with rare diseases. A notable example is the c.2991+1655 A > G variant in the *CEP290* gene, which is frequently observed in patients with Leber congenital amaurosis type 10 in Europe and North America[18]. Currently, Sepofarsen, a splice-switching ASO, designed to target this particular variant is under development[17,19].

We developed software, juncmut (https://github.com/ncc-gap/juncmut), which can identify SSCVs using only transcriptome sequence data. This method is finely tuned to remove most false-positives. Furthermore, it can be executed on a single transcriptome without the need for specifying groups during the analysis, allowing for the efficient acquisition of SSCV collection through unified processing of massive transcriptome sequences. After demonstrating the high precision and decent sensitivity of juncmut, we developed a registry of SSCVs (SSCV DB, https://sscvdb.io/) by applying juncmut to over 300,000 transcriptome sequence data from the SRA, which will be beneficial for gaining novel biological insight and finding new targets for splice-switching ASOs (the study design is depicted in Supplementary fig. 1).

## Results

### Method overview

In this work, we focus on the variants that form a novel splice donor/acceptor motif and generate a new splicing junction at that location. We exclude from consideration variants that are distant from the novel splice-site and act by modifying the efficacy of splicing enhancers. Additionally, we do not include variants that disrupt the normally used splice-sites, leading to the use of preexisting cryptic splice-sites. To develop a methodology for identifying SSCVs from transcriptome sequence data, we focused on the following properties:

1. SSCVs, particularly those associated with diseases, are rare in the population. Thus, the corresponding splicing junctions are typically not observed in the general population.
2. Mismatch bases corresponding to SSCVs are often observed in the short reads of transcriptome sequence data.

The latter is somewhat counterintuitive. It is generally considered that SSCVs, typically located on the intronic side of a novel splice-site, are expected to be spliced out, rendering them undetectable in short reads. However, owing to the heterogeneity of splicing effects and an imperfect penetrance of SSCVs (probably due to competition with existing splice-sites), mismatch bases showing SSCVs appear to be frequently observed in the transcriptome in many cases (Fig. 1b).

Following the step of transcriptome sequencing alignment, the overview of juncmut is as follows (Fig. 1c; see the "Methods" section for details). First, it extracts the set of rare splicing junctions (defined as infrequently observed in the Genotype-Tissue Expression project (GTEx)[20] transcriptome data), restricting those spanning annotated exon-intron boundaries (hereafter referred to as matching splice-site (SS)) to unannotated boundaries (primary novel splice-site (SS)). Next, juncmut lists all the possible variants that make the sequence at the primary novel SS closer to the consensus donor/acceptor motif (donor: AG|GTRAGT, acceptor: YYNYAG|R ('|' represents the exon-intron boundary)) compared to the reference genome sequence. Subsequently, juncmut checks whether they are observed in the

transcriptome, which would indicate a candidate for an SSCV. Then, all the short reads around the candidate SSCV are realigned to the abnormal transcripts predicted from the splicing junction as well as normal transcript sequences to confirm that the observed SSCV is not an alignment artifact. Finally, various filters, such as removing common variants (those whose allele frequencies are greater than 0.01 according to the gnomAD database[21]), are applied.

The term "primary" in the "primary novel SS" signifies the direct formation of a novel SS by an SSCV, particularly considering situations where an SSCV within a deep intronic region generates a cryptic exon and subsequently leads to the formation of another novel SS (referred to as secondary SS, as shown in the right panel of Fig. 1a).

### Evaluation of Juncmut approach

We evaluated the juncmut approach using the 1000 Genomes Project RNA sequencing data[8]. Among 645 transcriptome sequence data with matched whole-genome sequencing data available (445 individuals), we identified 153 SSCVs (90 distinct SSCVs, counting only once the variants sharing the same position and substitution identified across multiple samples). Most of the SSCVs (152 out of 153) were also detected as genomic variants by an investigation of raw whole-genome sequencing data[22], confirming that these SSCVs were indeed actual genomic-level variants (Fig. 1d; Supplementary Data 1). To measure the effect of splicing by these SSCVs, we utilized an independent dataset comprising 479 GTEx whole-genome sequences and their transcriptome sequences (8656 across 53 different tissues). For 24 distinct SSCVs in which at least one individual in GTEx had genomic variants in the whole-genome sequences, we computed the *p*-value measuring the difference in abnormal splicing ratio between SSCV-positive and -negative samples for each tissue and integrated these *p*-values across tissues. For most of the SSCVs (23 SSCVs), the combined p-values were highly significant (p-value ≤ $10^{-8}$), indicating that the SSCVs identified by juncmut have a strong impact on abnormal splicing (Figs. 1e, f, Supplementary figs. 2 and 3). Furthermore, compared with SSCVs identified by juncmut, the variant chosen by SpliceAI[23], a machine learning-based splicing effect predictor, showed modest significance (Fig. 1e, see "Methods" section for details). This is likely because juncmut, which directly confirms both genomic variation and splicing abnormalities from actual transcriptome sequence data, may be advantageous over purely prediction-based approaches from nucleotide sequences. We also compared another approach, AbSplice[24], which showed a similar trend to SpliceAI (see Supplementary fig. 4). In addition, we performed a comparison with our previous approach, SAVNet[3], which collects splicing-associated variants using paired genome and transcriptome data. Despite using only transcriptome data, juncmut detected approximately 33.3% (55 out of 165) of the SSCVs (restricted to those with ≤ 0.01 allele frequency) identified by SAVNet (Supplementary fig. 5a).

We also applied juncmut to 11,373 TCGA transcriptome sequence data, and identified 2104 SSCVs (Supplementary Data 2). For the 234 SSCVs identified from 847 transcriptomes with matched whole-genome sequencing data, we classified these SSCVs into germline (181), somatic (41), somatic or germline (2), ambiguous (1), and false-positive (9), based on the number of supporting reads as well as sequencing depths at the SSCV positions (Supplementary fig. 6). Consequently, the estimated false-positive ratio in terms of genomic variant status was 3.86% (9 out of 233). We also performed a comparison with SAVNet for this dataset, where juncmut was able to detect approximately 23.1% (25 out of 108) of the somatic SSCVs identified by SAVNet (Supplementary fig. 5b)[25]. The slightly lower sensitivity compared to the 1000 Genomes Project dataset may be attributed to the fact that the benchmark variant set was limited to somatic variants, which often have lower variant allele frequencies and smaller splicing changes than germline variants. Overall, juncmut achieves a certain

level of sensitivity and a high rate of precision, even though it uses only transcriptome data.

Furthermore, juncmut identified 69 distinct SSCVs affecting known cancer genes[26], including *ERCC2, TP53* (3 distinct SSCVs), *BCOR, CDK4, CHD8, CIC, FAT1, KMT2D, PIK3R1, PTEN,* and *SMAD4* (2 distinct SSCVs) (Supplementary fig. 7, Supplementary Data 3), some of which (*BCOR* p.R1375W, *PIK3R1* c.1426-13 A > G, and *PIK3R1* c.1426-8 T > A) were validated via a mini-gene assay (Supplementary fig. 8). As a prominent example, in the transcriptome of an adrenocortical carcinoma (ACC) sample, we detected an SSCV in a deep intronic region affecting *PRKAR1A*, a gene linked to ACC tumorigenesis[27]. This SSCV (c.892-129 C > G) created a novel 69 bp exon (the right of Fig. 1b), which is typically challenging to detect without whole-genome analysis. These results indicate that the juncmut approach can effectively catalog disease-associated variants.

### Screening of splice-site creating variants using the Sequencing Read Archive

Next, to acquire a more comprehensive list of SSCVs, we applied this approach to publicly available transcriptome sequence data from the SRA, mainly using the National Institute of Genetics (NIG) Supercomputer (Supplementary fig. 9)[13]. After the selection and quality checks (see "Methods"), we eventually processed 310,699 transcriptome sequence data (counted on the basis of run IDs), and integrated the results of the TCGA data. Most sequence data had either no SSCVs or just one, whereas sequences with higher base counts presented an increased likelihood of SSCV detection (Fig. 2a). Similar to the previous project[13], we basically focused on "distinct SSCVs," in which the variants with the same genomic position and substitution detected from multiple sequencing data were considered as one. We identified a total of 30,130 distinct SSCVs (21,219 donor and 8911 acceptor creating variants). Among these, 28,225 were in coding genes. Additionally, 18,357 (60.9%) were not registered in the gnomAD database (Supplementary fig. 10). As expected, the novel SSs created by SSCVs increased MaxEnt scores, making them comparable to the hijacked SSs (originally utilized splice-sites that were "hijacked" by the SSCVs; see Supplementary fig. 11).

The donor-creating SSCVs were most frequently found at the last exonic bases (the -1 position in Fig. 2b and Supplementary fig. 12). This may be because SSCVs on the exonic side of a primary novel SS, which do not disappear from RNA sequence after splicing, are detected with higher sensitivity compared to those on the intronic side. Compared with variants that create splice donor sites, those that form splice acceptor sites were more concentrated near exon-intron boundaries, indicating a stronger positional constraint for splice acceptor sites that require polypyridine tracts and branch points (Fig. 2c). A pronounced tendency was observed for the positional differences between primary novel SSs and hijacked SSs to be multiples of three, as also implied in a previous study[28]. This pattern might be attributable to the reduced susceptibility to nonsense-mediated decay (NMD) at these positions, as those SSCVs generate in-frame alterations usually without generating premature termination codons (PTCs), which in turn makes the splicing changes more detectable in transcriptome sequence data. This finding was corroborated by the observation that, in non-coding genes (which are not subject to NMD), the proportion of SSCVs whose shift size is a multiple of three decreased to approximately one-third (Fig. 2d). There were notable increases in the frequencies of donor-creating SSCVs whose relative positions of primary novel SSs to hijacked SSs are -4 and 5, attributable to the characteristics of the original splice donor sequence, as previously shown with smaller sets of SSCVs (Fig. 2e)[29].

### Splicing consequence classification of SSCVs

We developed a workflow for classifying SSCVs based on transcription consequence categories by processing the coverage and splicing junctions around SSCVs and their positional relationships with proximal exons (Fig. 1a, see "Methods" section for details). SSCVs were categorized into three classes: 12,018 with partial exon loss, 5105 with exon extension, and 7082 with cryptic exon inclusion. Additionally, 5925 SSCVs that were not classified into the three categories were set as ambiguous (Fig. 3a). In the case of cryptic exon inclusion, the median size of new exons was 102 bp (with an interquartile range (IQR) of 63 bp), which was slightly smaller compared to ordinary exon sizes (median: 123 bp) (Supplementary fig. 13).

Most frameshift transcripts generated by SSCVs harbored PTCs, many of which were predicted to be susceptible to NMD by the 50 bp rule[30]. In fact, 20.87% of in-frame transcripts generated by SSCVs possessed PTCs, especially for those appending additional bases to transcripts by exon extension and cryptic exon inclusion (Fig. 3b). The ratio of generating a PTC naturally increased with the larger size of the newly added exonic bases, probably due to the increased likelihood of encountering PTC-generating trinucleotides (Fig. 3c). Additionally, even in SSCVs that cause in-frame partial exon loss, 36 instances produced PTCs. This includes 21 cases where an SSCV creates a stop codon in the original transcript on the exon side of the primary novel SS, thereby transferring the stop codon to the new transcript. In the other 15 instances, changes in the combination of trinucleotides constituting codons near the splicing junction led to new stop codon formation. When classifying SSCVs in the coding region based on annotations that assume no splicing abnormalities (apparently 3044 silent, 6501 missense, and 559 nonsense variants), we identified 1611 SSCVs that, although seemingly silent, actually lead to protein truncation via the production of PTCs. Furthermore, 201 SSCVs, typically annotated as nonsense, actually prevent truncation by producing transcripts without PTCs[31] (Fig. 3d).

### On the association between splice-site creating variants and Alu exonization

Alu elements account for approximately 10% of the human genome and are the most abundant repeat elements. Typical size of Alu elements is about 300 bp, and they consist of two similar monomers, left and right arms (ancestrally derived from the 7SL RNA[32]), which are separated by an A-rich linker, followed by a poly-A tail (Supplementary fig. 14)[33,34]. Alu elements inserted in the intronic region play important roles in "exonization" by further acquiring mutations enhancing splicing[35–37]. Although this exonization has been studied mostly in the context of primate genome evolution[38], we have previously demonstrated that this process could be observed in cancer genome evolution[25]. Here, using a richer set of SSCVs collected in this study, we explored higher resolution relationships between Alu elements and SSCVs.

A total of 3102 SSCVs were overlapping with Alu elements with lengths of ≥ 200 bp (Supplementary Data 4); most were located in deep intronic regions. Among these, 1132 overlapped with Alu elements in the sense direction (983 donor creations, and 149 acceptor creations), whereas 1970 overlapped in the antisense direction (1631 donor creations, and 339 acceptor creations). Next, we investigated the positions of the primary novel SSs formed by these SSCVs relative to the coordinate of a reference Alu sequence (Supplementary fig. 14)[39]. We identified 14 prominent clusters (each with more than 50 SSCVs), four of which (at the 102nd, 125th, 225th, and 268th nucleotides in the reference Alu sequence) created splice donor sites in Alu-sense sequences (hereafter referred to as "Alu-sense donor clusters"). Eight clusters (at the 22nd, 37th, 64th, 137th, 157th, 173rd, 199th, and 205th nucleotides) were classified as "Alu-antisense donor clusters," and two (at the 276th and 280th nucleotides) were "Alu-antisense acceptor clusters" (Fig. 4a). There were no prominent "Alu-sense acceptor clusters." While these clusters were partly identified through previous analyzes[38], our rich SSCV datasets have revealed signals of hotspots with unprecedented prominence. The SSCVs associated with the most prominent 102nd base Alu-sense donor cluster were

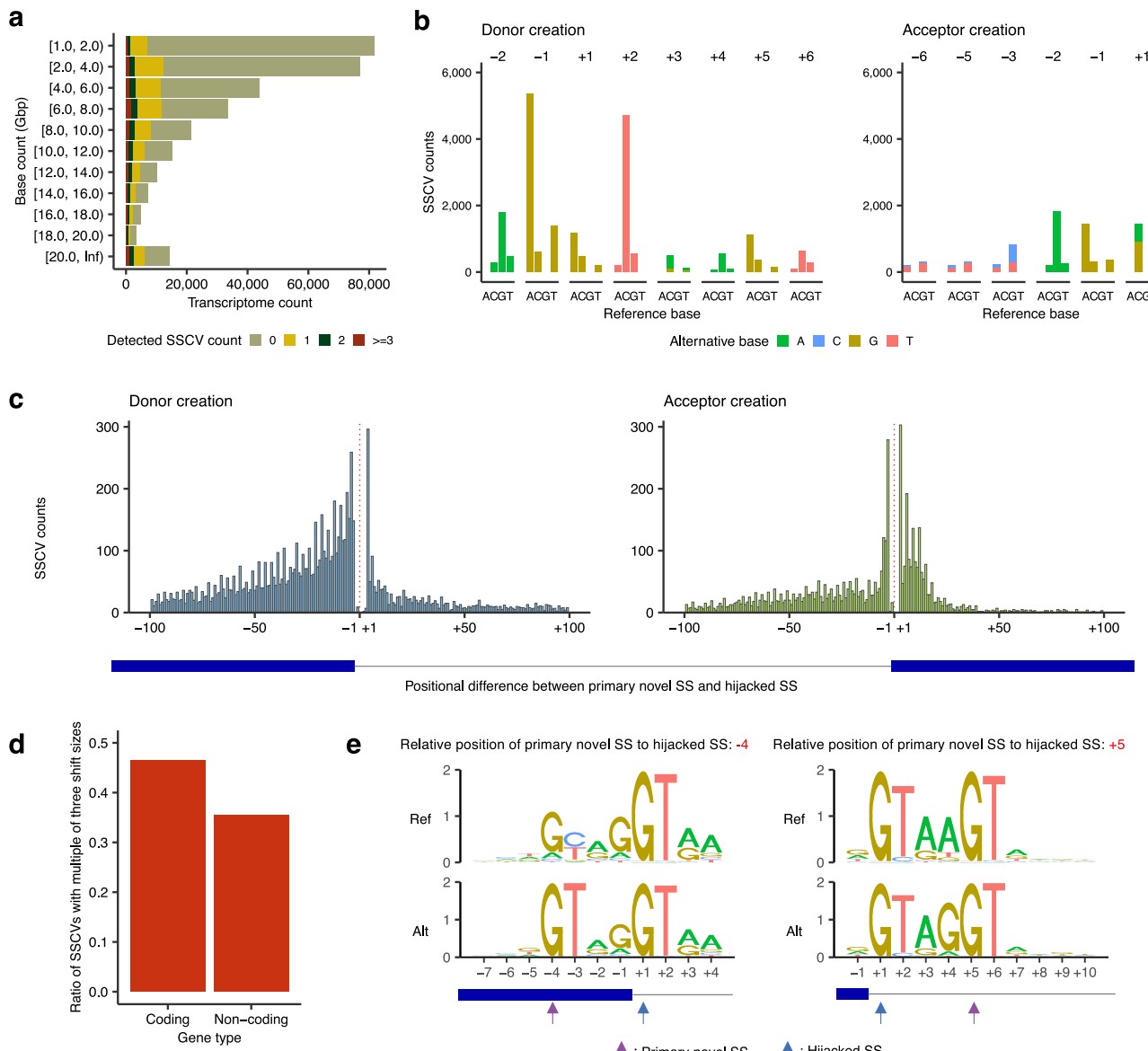

**Fig. 2 | Overview of SSCVs identified from Sequencing Read Archive and The Cancer Genome Atlas. a** Frequencies of transcriptome sequence data analyzed, binned by the amount of base counts. Transcriptome data were also grouped by the number of detected SSCVs. For example, in the rows [1.0, 2.0), whose base counts are equal to or more than 1.0 Gbp and less than 2.0 Gbp, zero, one, two, three or more SSCVs were identified in 74,805, 5464, 913 and 426 transcriptome sequence data, respectively. **b** Base substitution patterns of SSCVs according to their relative position to primary novel SSs. Different colors are used to display different types of alternative bases. The x-axes represent different reference bases, and the y-axes represent the numbers of variants. **c** Histogram showing the distribution of relative position of primary novel SSs to their hijacked SSs (original SSs) for donor (left) and acceptor (right) creating SSCVs. Red dashed lines represent exon-intron boundaries. **d** Fraction of SSCVs with multiples of three shift sizes (difference between primary novel SSs and hijacked SSs) stratified by coding and non-coding genes. **e** Sequence motifs of SSCVs with the relative position of primary novel SS to hijacked SS is -4 (left) and +5 (right), respectively. The "GT" dinucleotides at the intrinsic intron edge endow the -4 bp position with the potential to form a novel donor site, featuring "GT" at the fifth and sixth positions within the new intron. In addition, the inherent intron's fifth and sixth base pairs often comprise "GT" at the donor site, this configuration frequently corresponding to the first two intronic bases of a novel splice donor at the +5 bp position. **a**, **b**, **c**, **d**, **e** Source data are provided as a Source Data file.

predominantly located in AluJ sequences. This is probably due to the characteristics of AluJ sequences in those regions (Supplementary fig. 15). On the other hand, the majority of events in the other clusters occurred on AluS sequences, likely reflecting the abundance of sequences classified as AluS in the human genome[38]. Through motif analysis, we obtained a comprehensive landscape of Alu exonization mutation patterns for each cluster (Supplementary figs. 16–18), which includes the mutation pattern in major hotspots, such as the novel acceptor creation at the 280th position due to the mutation of a G base at the 283rd position (3rd intronic base from the primary novel SS) to C or T (Supplementary fig. 18)[36].

SSCVs in Alu-antisense sequences tend to result in cryptic exon inclusions, and most of these novel exons were confined to the Alu sequences (Fig. 4b). Splice donor sites, co-created with splice acceptors directly formed by SSCVs, usually matched the Alu-antisense donor clusters. Similarly, the reverse was true, with secondarily created splice acceptor sites matching with the Alu-antisense acceptor clusters (Supplementary fig. 19). The landscape of the exonized part of Alu sequence by SSCVs revealed that most new exons were confined to either the left arm or the right arm, with the latter being more frequent (Fig. 4d). On the other hand, SSCVs in Alu-sense sequences were significantly less likely to form new exons;

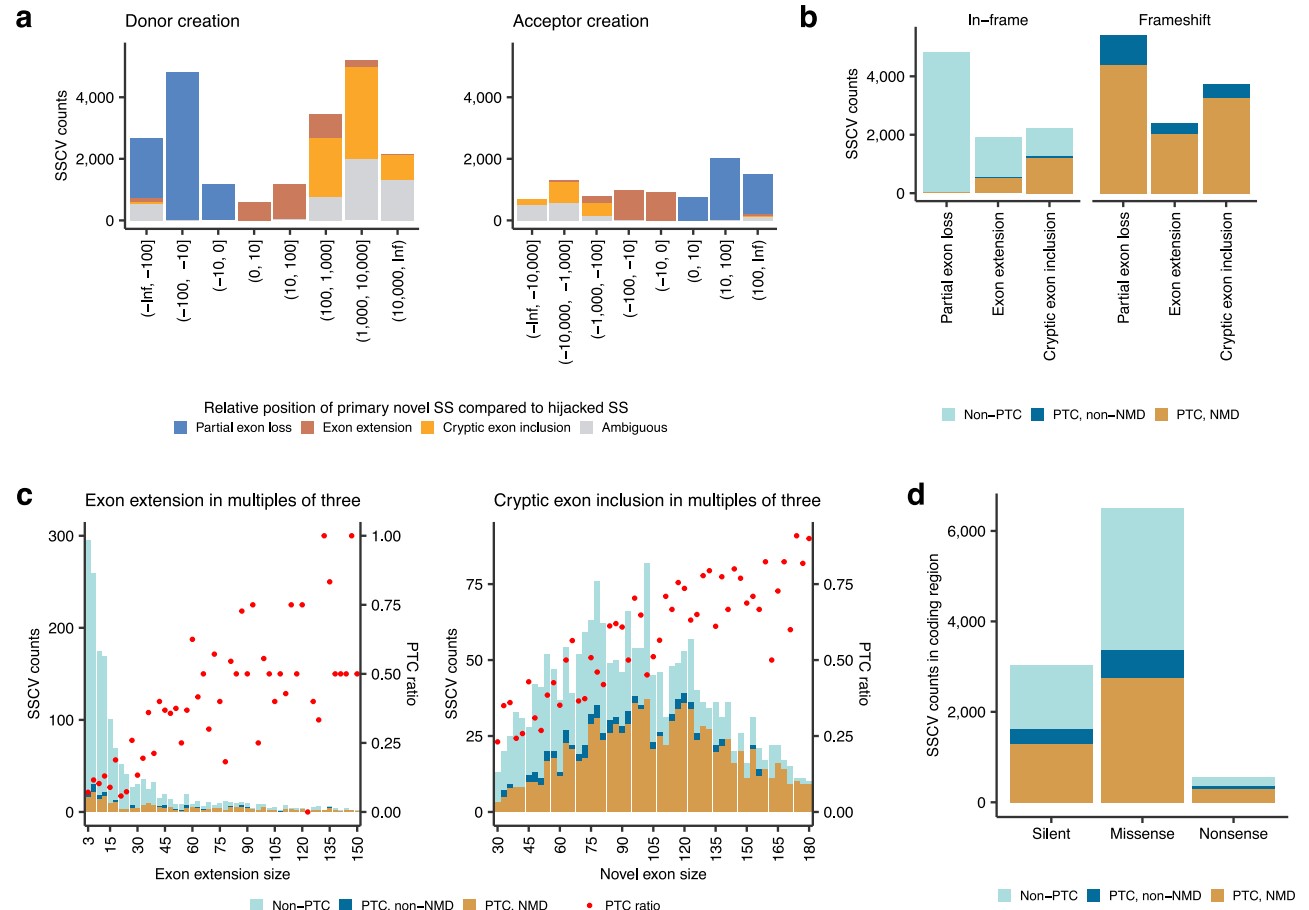

**Fig. 3 | Classification and summary of SSCVs based on splicing consequences.**
**a** Counts of distinct SSCVs creating novel donor (left) and acceptor (right) sites stratified by splicing consequences at each relative position of primary novel SSs compared to hijacked SSs. **b** Counts of distinct SSCVs leading to in-frame (left) and frameshift (right) partial exon loss, exon extension, and cryptic exon inclusion, stratified by PTC generation and NMD susceptibility. **c** Counts of distinct SSCVs at each size of augmented exon (restricted to multiples of three), for both exon

extension and cryptic exon inclusion, stratified by PTC generation and NMD susceptibility. Each red point represents the ratio of PTC generation. **d** Counts of distinct SSCVs located in coding regions, categorized by mutation type assuming no abnormal splicing (silent, missense, and nonsense). These counts are further stratified by PTC generation and NMD susceptibility. **a**, **b**, **d** Source data are provided as a Source Data file.

even when they were created, they tended to extend beyond the Alu sequences (Fig. 4c).

Finally, when we aligned the sequences of the left and right arms, which are separated by an A-rich linker, we found that three Alu-antisense donor clusters on the left arm (at the 22nd, 37th, and 64th nucleotides) aligned with clusters on the right arm (at the 157th, 173rd, and 199th nucleotides, respectively). This finding indicates that the splice donor site hotspots of Alu-antisense exonization have been conserved throughout the evolution of the Alu sequence from its precursor, the 7SL RNA (Fig. 4e).

**Splice-site creating variants affecting disease-related genes**
To identify SSCVs linked to disease, we extracted SSCVs that affect genes associated with disease (genes with at least one registered pathogenic variant in ClinVar[40] or listed in the Cancer Gene Census[41], see the "Methods" section for details). In total, 5121 SSCVs affected disease-related genes (826 were on cancer-related genes and 132 were on the ACMG SF list[42]): 2248 were categorized as partial exon loss, 897 as exon extension, 1222 as cryptic exon inclusion, and 754 as ambiguous based on splicing consequences (Fig. 5a). Of these, only 82 SSCVs were registered as pathogenic in ClinVar, indicating that most of the SSCVs were potentially new disease-related variants. Genes with a greater number of exons tended to have more SSCVs detected (Supplementary fig. 20).

The top frequent gene was *ATM,* which encodes a protein crucial for the DNA damage response and cell cycle regulation, and mutations in this gene cause ataxia-telangiectasia, a disorder characterized by neurological degeneration[43]. In total, 22 SSCVs were identified in *ATM* (Fig. 5b). Among the 21 SSCVs with non-ambiguous splicing consequences, 20 were predicted to generate PTCs. Six of these SSCVs have already been registered as pathogenic in the ClinVar database, five of which were variants in deep introns, probably because *ATM* is being actively studied[16]. However, this indicates that our approach can successfully detect known pathogenic SSCVs and also suggests the potential pathogenicity of the other SSCVs.

We would also like to highlight *CREBBP,* a multifunctional transcriptional coactivator with intrinsic acetyltransferase activity. The most common mutations in *CREBBP* are missense mutations, which predominantly occur in the lysine acetyltransferase (KAT) domain, especially at positions R1446, Y1450, Y1483, and Y1503[44]. Among the 10 SSCVs, four were located in the KAT domain, all of which resulted in in-frame transcripts without introducing PTCs and were predicted to generate protein changes of E1459_L1464del, K1462_V1467del, G1465_Y1466insX[3], and E1576_G1577insX[34] (Fig. 5c, Supplementary fig. 21). Therefore, the SSCV profile might reflect a selective pressure related to the KAT domain, similar to the profile of somatic SNVs. Furthermore, the closely related paralog, *EP300,* also exhibited multiple SSCVs resulting in in-frame, non-PTC generating transcripts in the KAT domain (Supplementary fig. 22).

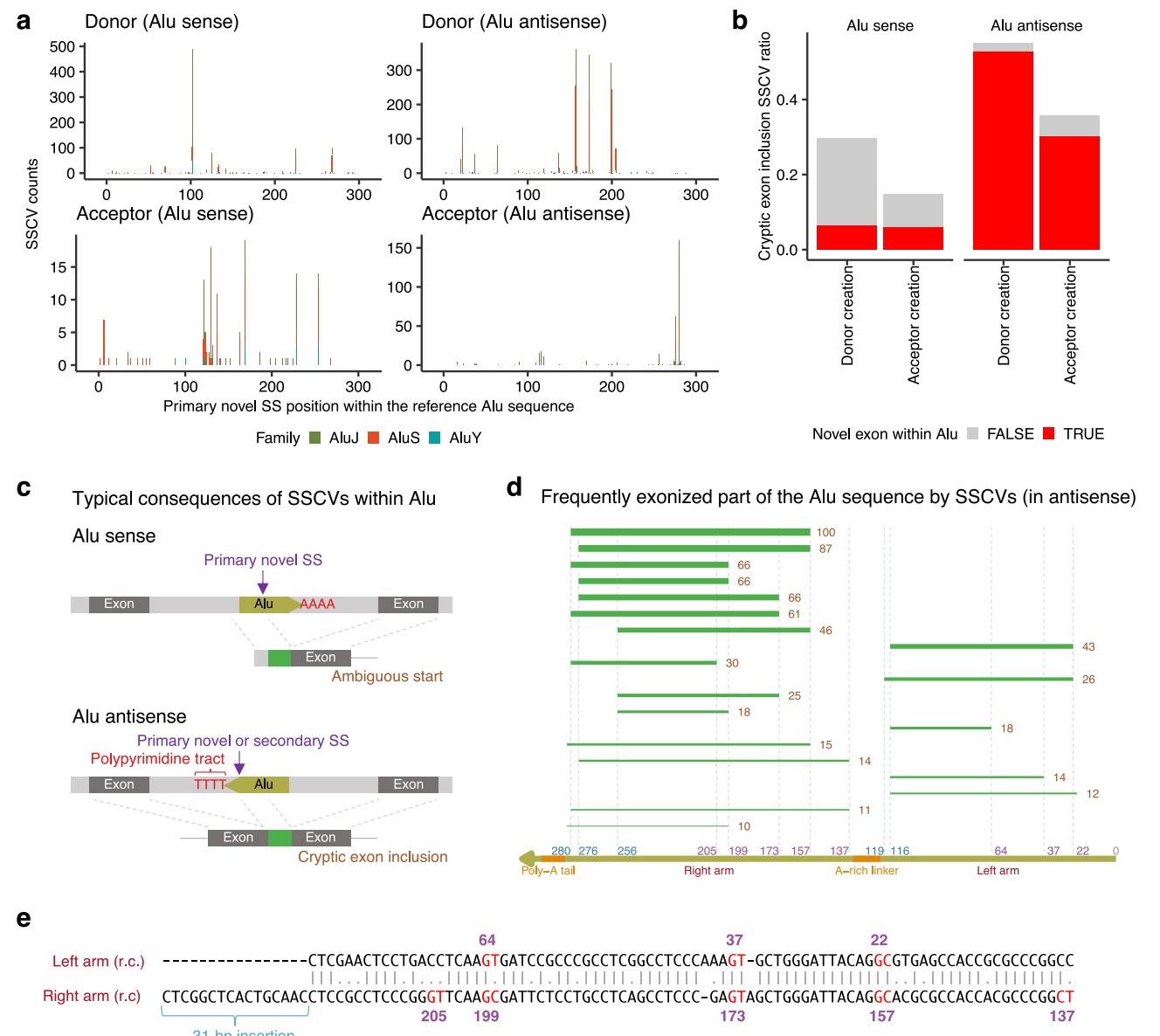

**Fig. 4 | Characteristics of Alu exonization by SSCVs. a** Counts of distinct SSCVs within Alu sequences, at each primary novel SS mapped to the reference Alu sequence coordinates. The counts are stratified by Alu family (AluJ, AluS, and AluY). These counts are faceted by the creation of donor and acceptor sites, and the orientation of the Alu sequences relative to transcripts (sense and antisense). **b** The ratio of SSCVs forming novel exons (classified as cryptic exon inclusion by splicing consequence) at each motif creation type (donor and acceptor), and in Alu sequence orientations (sense and antisense). These ratios are further stratified based on whether the novel exons are confined within Alu sequences or not. **c** Typical splicing consequences of SSCVs within Alu sequences. SSCVs located on sense-inserted Alu sequences do not form exons and may create novel transcription start sites in an ambiguous manner. Conversely, SSCVs on antisense-inserted Alu sequences are likely to form novel exons within the Alu sequences. **d** Frequently exonized parts by SSCVs in antisense-inserted Alu sequences. The green lines indicate the exonized parts, and the numbers on the right represent the counts observed in this study. Additionally, the thickness of these green lines corresponds to the frequency. **e** Pairwise alignment of the Alu reference sub-sequences (reverse complemented) containing the Alu-antisense donor clusters in the left arm and right arm. It is observed that the 22nd nucleotide corresponds with the 157th, the 37th with the 173rd, and the 64th with the 199th. The term 'r.c'. stands for reverse complement. **a**, **b** Source data are provided as a Source Data file.

*TP53* is the most frequently mutated gene in human cancer, and 80% of *TP53* mutations are missense mutations in the DNA binding domain that are believed to have different functions than truncating mutations such as nonsense mutations[45]. We detected eight SSCVs in *TP53*, seven of which located in the DNA-binding domain (Fig. 5d). Notably, out of the three SSCVs commonly annotated as nonsense mutations, two (Y126* and E258*) actually generated in-frame transcripts without PTCs (Y126del and E258_S261del, respectively), rescuing the truncation. Meanwhile, of the four apparently missense SSCVs, one produced a transcript with a PTC. Although the complex splicing consequences of these *TP53* variants have not been widely studied, gaining a deeper understanding of them is crucial for tailoring treatments according to specific mutation categories[46].

**Gain-of-function deep intronic *NOTCH1* SSCVs: examples of novel SSCVs targetable by antisense oligonucleotides**

We searched for novel disease-related SSCVs targetable by splice-switching ASOs and detected deep intronic SSCVs in *NOTCH1* (c.5048-132 G > C, c.5048-132 G > T) in multiple samples. These SSCVs form splice acceptor sites 129 bp away from exon 28 in *NOTCH1*, producing

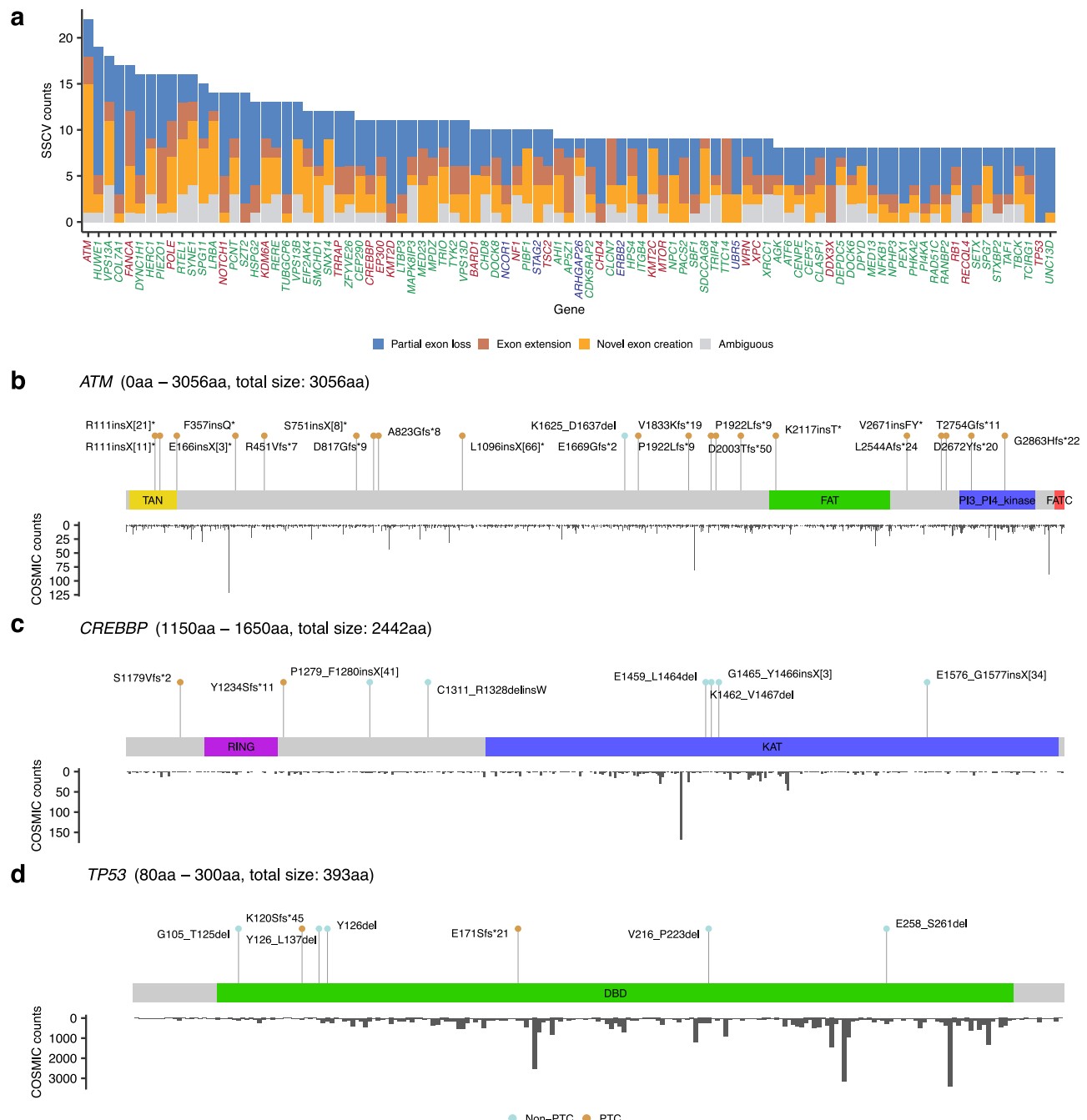

**Fig. 5 | SSCVs affecting disease-related genes. a** SSCV counts for each disease-related gene, stratified by splicing consequences. Genes registered in the Cancer Gene Census are colored blue, those registered as Pathogenic in ClinVar are colored green, and genes registered in both are colored red. **b**, **c**, **d** Lollipop plots of the detected SSCVs affecting *ATM* (**b**), *CREBBP* (**c**), and *TP53* (**d**). These plots display labels denoting protein changes based on splicing consequences (using HGVS notation), and points are colored according to PTC generation (light blue for non-PTC, brown for PTC). Each plot is accompanied by the counts of somatic mutations at each amino acid position, as recorded in COSMIC. For *CREBBP* and *TP53*, the display is focused on the KAT and DNA binding domains, respectively. See also Supplementary figs. 21 and 22, and SSCV DB (https://sscvdb.io). **a** Source data are provided as a Source Data file.

proteins that are 43 amino acids larger than usual (S1723delinsX[44]) (Fig. 6a). These SSCVs were detected in a total of nine cancer samples (breast cancer, metastatic prostate cancer, acute lymphoblastic leukemia, and lung cancer) in the TCGA and the SRA (see Supplementary fig. 23 and Supplementary Data 5). In fact, internal tandem duplication around exon 28 in *NOTCH1*, which is frequently observed in T-cell acute lymphoblastic leukemia patients, is known to cause ligand-independent NOTCH1 activation by expanding the extracellular juxtamembrane region[47,48]. Since the transcripts induced by the SSCVs

were expected to elongate the same juxtamembrane region, we considered the possibility that these SSCVs could cause similar activation (Fig. 6b).

To conduct a detailed functional analysis of SSCVs, we established CRISPR genome editing models derived from PC-9[49,50], a lung cancer cell line, for each SSCV (c.5048-132 G > C and c.5048-132 G > T). We successfully introduced the *NOTCH1* deep intronic SSCV into the cell line, confirming a 129 bp exon extension (Fig. 6c, Supplementary fig. 24). Next, we assessed the expression levels of the

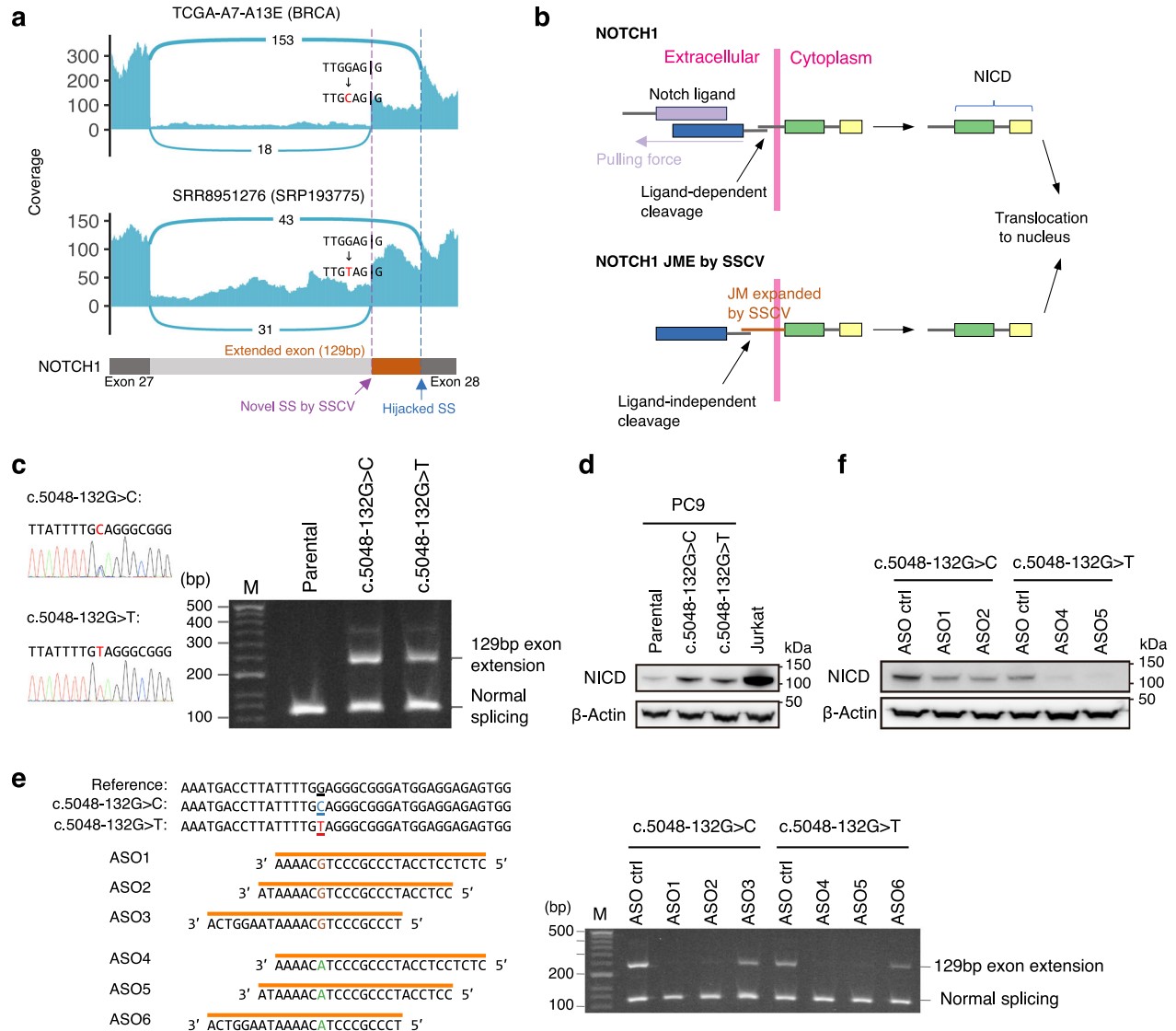

**Fig. 6 | Gain-of-function SSCVs in NOTCH1 and their inhibition by splice-switching antisense oligonucleotides. a** Sashimi plot for samples with *NOTCH1* c.5048-132 G > C (TCGA-A7-A13E, upper) and c.5048−132 G > T (SRR8951275, lower) mutations. These mutations were expected to result in a 129 bp exon extension (without any stop codon within it), leading to the production of a protein with an additional 43 amino acids. **b** Predicted schematics of the mechanisms for ligand-independent cleavage of NOTCH1 juxtamembrane expansion (JME) induced by the SSCVs. **c** (left) Sequencing chromatograms of two *NOTCH1* DNA derived from single clones of two CRISPR-edited PC-9 cell lines (c.5048-132 G > C and c.5048-132 G > T). (right) The PCR amplicons spanning *NOTCH1* exon 27 and exon 28 show a 129 bp exon extension in clones with the indicated *NOTCH1* genotype. 'M' in the lane stands for the 100 bp marker. **d** Western blot analysis of the NOTCH intracellular

domain (NICD) in CRISPR-edited clones. As a positive control, analysis is also provided on the Jurkat cell line, which is known to have an internal tandem duplication in exon 28, resulting in the insertion of 17 amino acids in the extracellular juxtamembrane domain. **e** (left) Schema depicting the design of splice-switching ASOs targeting c.5048-132 G > C (ASO1, ASO2, and ASO3) and c.5048-132 G > T (ASO4, ASO5, and ASO6). (right) Images of PCR amplicons spanning *NOTCH1* exon 27 and exon 28 generated from the cDNA of CRISPR-edited clones treated with indicated ASOs for two days. 'Ctrl' stands for control. **f** Western blot analysis of the NOTCH intracellular domain (NICD) in CRISPR-edited clones treated with indicated ASOs for three days. All experiments have been performed in at least two independent experiments. **c**, **d**, **e**, **f** Source data are provided as a Source Data file.

NOTCH intracellular domain (NICD), a marker of activation, via Western blot analysis. This evaluation revealed that the SSCV provoked substantial activation (Fig. 6d). Finally, to further examine potential interventions, we designed mutant-selective ASOs with full phosphorothioate (PS) + 2'-O-Methoxyethyl (2'MOE) modifications, and treated the genome editing models. Following the administration of these ASOs, we observed the disappearance of the 129 bp exon extensions caused by the SSCVs (Fig. 6e) and a suppression of NOTCH1 activation, as evidenced by decreased NICD expression (Fig. 6f). These findings demonstrate that these SSCVs lead to the activation of NOTCH1, which can be suppressed by splice-switching ASOs.

## Discussion

We have developed an innovative approach for cataloging splice-site creating variants (SSCVs) by leveraging the wealth of transcriptome data in public repositories. The extensive collection of SSCVs obtained enabled us to gain various biological insights, including the identification of prominent hotspots in Alu exonization and the heterogeneous effects of SSCVs beyond loss-of-function. Importantly, we identified gain-of-function SSCVs in the deep intronic region of *NOTCH1*, and demonstrated that its activity can be suppressed by ASOs. Further scrutiny of our SSCV catalog will lead to the identification of more ASO-amenable targets, enhancing personalized splice-switching ASO therapies[16,51,52].

A limitation of our approach is inherent biases, and low sensitivity compared to the approach using both genome and transcriptome. This is because juncmut relies on a limited number of short reads with mismatches corresponding to SSCVs, arising from the heterogeneity of splicing abnormalities in transcriptome sequence. When exploring splicing-associated variants in individual patients for clinical purposes, we recommend a combination of juncmut and other approaches involving both genome and transcriptome sequence data[3,23,53]. However, the ability to acquire a catalog of SSCVs through reanalysis of existing transcriptome sequence data is an attractive feature. Particularly because juncmut can be performed on individual transcriptome sequence data, execution on large-scale transcriptome sequences is highly convenient.

Our saturation analysis indicates that the continuous application of this method will lead to the identification of an increasing number of SSCVs as more sequence data is incorporated into the repository (Supplementary fig. 25). The next important challenge will be to become capable of systematically and accurately predicting variants responsible for rare diseases and cancers from a vast list of SSCVs. SSCVs exhibit heterogeneous effects, with loss-of-function and gain-of-function variants intermingling for each gene, which makes this prediction challenging. However, by overcoming this, we can develop a system that autonomously archives important disease-related variants, some of which are targetable by splice-switching ASOs.

## Methods

### Selection of transcriptome sequence data from the sequence read archive

We selected public transcriptome sequence data from the Sequence Read Archive (SRA) in a manner similar to a previous study[13]. We queried the SRA website (https://www.ncbi.nlm.nih.gov/sra) using the following search terms: "platform illumina"[Properties] AND "strategy rna seq"[Properties] AND "human"[Organism] AND "cluster public"[Properties] AND "biomol rna"[Properties]. We then extracted samples with a base number of ≥1 billion to ensure sufficient sequence coverage for reliable mutation detection. We removed run data that could not be downloaded even after repeated attempts (likely due to technical issues). We also excluded sequence data that had severe issues, such as inconsistencies between the two paired-end files, discrepancies between sequence letters and base qualities, etc. Furthermore, we discarded run data with an extremely high number of SSCVs, attributable to potential DNA contamination and other factors.

### Downloading sequence data

We utilized the SRA Toolkit version 2.11.0. Initially, we executed the 'prefetch' command with the '–max-size 100000000' option to download the SRA format file. Subsequently, we used the 'fasterq-dump' command with the options '-v –split-files.'

### Alignment of transcriptome sequence data

We used the GRCh38-based reference genome provided by the Genomic Data Commons (https://gdc.cancer.gov/about-data/gdc-data-processing/gdc-reference-files). First, genome indexes were generated using STAR version 2.7.9b[54] with this reference genome and the release 31 GTF file from GENCODE, employing the '–sjdbOverhang 100' option. For each sample, alignment to the reference genomes was conducted using the same version of STAR with the following options: '–runThreadN 6 –outSAMtype BAM Unsorted –outSAMstrandField intronMotif –outSAMunmapped Within –outSJfilterCountUniqueMin 1 1 1 1 –outSJfilterCountTotalMin 1 1 1 1 –outSJfilterOverhangMin 12 12 12 12 –outSJfilterDistToOtherSJmin 0 0 0 0 –alignIntronMax 500000 –alignMatesGapMax 500000 –alignSJstitchMismatchNmax -1 -1 -1 -1 –chimSegmentMin 12 –chimJunctionOverhangMin 12.' After alignment, BAM files were sorted, converted to CRAM format, and indexed using SAMtools version 1.9 (https://www.htslib.org/).

### A workflow for the discovery of splice-site creating variants via juncmut

**Generating splicing junction control panels.** We initiate the identification of aberrant splicing junctions within transcriptome sequence reads (more specifically, the SJ.out.tab files produced by STAR). A splicing junction is characterized by its chromosomal location, the start coordinate, and the end coordinate of the intron within each transcript. To detect "aberrant" splicing junctions, we established control panels of splicing junctions, which were consistently observed across multiple samples within specific cohorts. We processed transcriptome sequences from two cohorts, the Cancer Genome Atlas (TCGA) and the Genotype-Tissue Expression (GTEx) Project, and generated two control panels.

For the TCGA dataset, we processed 742 transcriptome sequence samples from non-tumorous tissues of cancer patients. We then identified a list of splicing junctions supported by a minimum of two reads across at least four different samples. In the case of GTEx, we analyzed 8656 transcriptome sequences (comprising 479 individuals across 53 different tissues). We extracted splicing junctions present in any tissue with two or more supporting reads in at least eight individuals.

**Detection of aberrant splicing junctions.** For each transcriptome sequence aligned by STAR, we parse the SJ.out.tab file. Here, we extract splicing junctions focusing on those possibly associated with novel splice-site creation by a mutation, where one edge matches within a 5 bp margin to the exon-intron boundary of a known transcript (GENCODE Comprehensive gene annotation Release 31), and the other edge does not correspond to any known exon-intron boundary (suggesting the formation of a novel splice-site). Additionally, we adjust the positions to ensure that one side of the edge perfectly aligns with the exon-intron boundary. The obtained splicing junctions at this stage are referred to as "primary novel splicing junctions (SJs)." For each primary novel SJ, we term the edge of a splicing junction that deviates from the exon-intron annotation as the "primary novel splice-site (SS)," and the edge that aligns with the annotation as the "matching splice-site (SS)."

We next remove primary novel SJs according to the following criteria:

- Primary novel SJs with fewer than three supporting reads are excluded.
- Primary novel SJs that are registered in the splicing junction control panels generated in the previous section are excluded.
- For each primary novel SJ, we count the total number of supporting reads of splicing junctions sharing the matching SS. If the supporting read count for primary novel SJ constitutes less than 5% of this total, it is excluded.
- For each primary novel SJ, we count the number of intersecting splicing junctions. If the number of intersecting splicing junctions is five or more, we remove this primary novel SJ.

**Identification of variants that account for the corresponding aberrant splicing junctions.** For each remaining primary novel SJ, we perform 'samtools mpileup' and search for mutations that can explain the formation of the associated primary novel SS.

For splice donor site creation (where the matching SS is an annotated acceptor), we focus on the two exonic bases (positions -2 and -1) and the six intronic bases (positions +1 to +6), relative to the primary novel SS. More specifically, we restrict our search to mutations that result in

- A, and G at positions -2, -1,
- G, T, A or G, A, G, and T at positions +1, +2, +3, +4, +5, and +6, respectively.

In addition, we require that the first two bp of the intron in the primary novel SS be 'GT' following the mutation.

For splice acceptor site creation (where the matching SS is an annotated donor), we focus on the six intronic bases (positions -6 to -1) and the one exonic base (position +1) relative to the primary novel SS. We search for mutations that result in

- C or T at positions -6, -5, and -3,
- A at position -2,
- G at position -1,
- A or G at position +1.

Additionally, we request that the last two bp of the intron in the primary novel SS be 'AG' following the mutation.

If at least two mismatch corresponding to the relevant mutation is detected from the short reads of transcriptome sequence data, it is set as the candidate for splice-site creating variants (SSCVs) and subjected to validation via the subsequent realignment procedure.

**Validation of candidate splice-site creating variants via short read realignment.** Special attention should be given to confirming that mismatch bases in the transcriptome sequence are not artifacts resulting from alignment errors[55]. Therefore, we perform an additional filtering step based on the realignment. For each candidate SSCV and its associated primary novel SJ, we prepare three types of "mini-transcripts":

- Primary novel transcript: extending 10 base pairs of the transcript sequence from both edges of the primary novel SJ.
- Reference transcripts: extending 10 base pairs of transcript sequences from both edges of the splicing junctions of know transcripts that share the matching SS (thus potentially resulting in multiple mini-transcripts).
- Intron retention transcripts: extending 10 base pairs of genome sequences in both directions from the position of the SSCV.

Furthermore, if the region of the transcript includes the SSCV, we also supply a version of the transcript with the SSCV mutation inserted. Consequently, we are able to generate at most six types of mini-transcripts: primary novel transcripts with and without the SSCV, reference transcripts with and without the SSCV, and intron retention transcripts with and without the SSCV.

Next, for each short read surrounding the position of the SSCV, we perform alignment on each type of mini-transcript using edlib[56] and extract the mini-transcript satisfying the following:

- The edit distance of the alignment must be two or less.
- There should be no mutations within 5 bp from the position of the candidate SSCVs.

Then, we choose the mini-transcript with the minimum edit distance. If there is a tie, we choose in the following order: reference transcript without the SSCV, intron retention transcript without the SSCV, primary novel transcript without the SSCV, reference transcript with the SSCV, intron retention transcript with the SSCV, and primary novel transcript with the SSCV.

If at least one read is classified as aligning with the primary novel, reference or intron-retention transcript with the SSCV, then the SSCV is retained as a final output.

## A workflow for classifying the consequences of splice-site creating variants

For each SSCV and its associated primary novel SJ detected by juncmut, we first identify a reference transcript. Then, based on this transcript, we classify the types of splicing consequences. Furthermore, we predict the resulting amino acid changes, the generation of premature termination codons (PTCs), and assess the susceptibility to nonsense-mediated decay (NMD).

**Identification of a reference transcript for the SSCV.** For each SSCV, the coordinate of the matching SS (the edge of the splicing junction that matches the exon-intron boundary of known transcripts) is identified from the corresponding primary novel SJ. Then, we extract all transcripts that possess this matching SS within their exon-intron boundaries. Subsequently, we determine the transcript based on the following priorities:

1. Registered as "MANE Select."
2. Registered as "MANE Plus Clinical."
3. The largest transcript (in the case of a tie, the transcript with the earlier ENST transcript ID is selected).

**Classification of the types of splicing consequences.** First, we identify the exon affected by the SSCV (referred to as the "affected exon"). In the context of donor site creation, the affected exon is defined as the one whose start position is closest to, yet before, the primary novel SS. The end position of this affected exon is termed the "hijacked SS." Conversely, for acceptor site creation, the affected exon is the one whose end position is closest to, but after, the primary novel SS, with the start position of this affected exon being the "hijacked SS."

The splicing consequences of the SSCV are classified as follows:

1. "Partial exon loss" if the primary novel SS is located within the affected exon.
2. "Cryptic exon" if the primary novel SS is located downstream or upstream (for donor and acceptor creation, respectively) of the affected exon, and if there is a splicing junction with one edge corresponding to the hijacked SS and the other edge within 300 bp upstream or downstream (for donor and acceptor creation, respectively) of the primary novel SS.
3. "Exon extension" if the SSCV is not classified as a "cryptic exon," and there is a sequence depth of one or greater observed from the hijacked SS to the primary novel SS.
4. "Ambiguous" for all other cases.

**Predicting the resulting amino acid changes and susceptibility to nonsense-mediated decay.** For SSCVs predicted to result in "partial exon loss," "cryptic exon," or "exon extension," we investigate the consequent protein changes, determine whether they are in-frame or not, assess the generation of PTCs, and predict the susceptibility to NMD.

First, we verify whether the primary novel SJ is completely contained within the 5'UTR or the 3'UTR, and we exclude those scenarios from further analysis. We also ignore the cases where SSCVs cause skipping of the start or stop codon.

With respect to the changes in the transcript, we compare the size changes from the reference transcript to determine whether they are a multiple of three (in-frame) or not (frameshift). We then extract the predicted amino acid sequence to assess whether a PTC occurs before the original stop codon position (resulting in a PTC), or the original stop codon remains unchanged without generating a PTC (non-PTC). Furthermore, in cases where a PTC is formed, we check whether this PTC is located less than 50 bp upstream of the last exon-intron junction (the 50 bp rule[30]).

## Evaluation of juncmut using 1000 Genomes Project whole genome and transcriptome sequence data
**Validation of mutation status of splice-site creating variant detected by juncmut using whole-genome sequencing.** For each SSCV, we performed 'samtools mpileup' directly on the corresponding CRAM file stored in Amazon Web Services (s3://1000genomes/1000G_2504_high_coverage/data/). We decided that the SSCV predicted from RNAseq is a genuine genomic mutation (although the effect on splicing is uncertain at this point) if more than two reads support the base corresponding to the SSCV, and the proportion of these supporting reads exceeds 5% of all reads covering that position.

**Validation of splice-site creating variants by juncmut using multi-tissue transcriptome sequence data.** We utilized GTEx data to verify whether SSCVs identified by juncmut in the 1000 Genomes Project transcriptome truly lead to significant splicing changes. We downloaded the GTEx transcriptome sequence data from the Sequence Read Archive and aligned them following the juncmut workflow. Also, we downloaded GTEx V7 whole genome genotype calls. For each SSCV, we counted the number of supporting reads for the corresponding primary novel SJ and hijacked SJ (splice junction connecting the matching SS and the hijacked SS in the reference transcript), and we calculated the ratio of the primary novel SJ (#primary novel SJ / (#primary novel SJ + #hijacked SJ)) by parsing the SJ.out.tab file. Then, for each tissue, we calculated the *p*-value that measures the difference in the ratio of the primary novel SJ between samples with and without the SSCV using a one-sided Wilcoxon rank-sum test with the wilcox.test function in the R language. We integrated these *p*-values using Fisher's method across tissues.

**Comparative evaluation of juncmut against SpliceAI and AbSplice.** We also evaluated variants predicted to cause splice-site activation via SpliceAI as a comparison to juncmut. For this, we downloaded VCF files from the 1000 Genomes Project from s3://1000genomes/ 1000G_2504_high_coverage/working/20201028_3202_phased/. Then, we added the SpliceAI score using the precomputed file for all SNVs and 1 base insertions, and 1–4 base deletions within genes. Next, we extracted SNVs with allele frequencies ≤ 0.01 satisfying the following criteria:

- SNVs possessed by either of the 445 individuals whose matched transcriptome sequence data are available.
- SNVs where SpliceAI Delta score for acceptor or donor gain (DS_AG or DS_DG) is equal to or above 0.1.

For each remaining SNV, we identified a novel splice-site (corresponding to primary novel SS in the juncmut) using the information on Delta positions (DP_AG or DP_DG) provided by the SpliceAI annotation. Then, based on GENCODE Basic annotation (Release 39), we chose the reference transcript, identified the matching SS and hijacked SS, and obtained the corresponding hijacked SJ. Finally, for variants that were called in at least one GTEx sample, we calculated the combined *p*-values across tissues as above.

For comparing AbSplice, we first downloaded precomputed SpliceMaps 'splicemap_download –version gtex_v8' command, provided in the package at https://github.com/gagneurlab/splicemap (commit: 9e9831f). Then we executed 'mmsplice_splicemap', 'spliceai', and 'absplice_dna' in turn, creating a conda environment using the contig file downloaded from https://github.com/gagneurlab/absplice (commit: 5d8bfb) and installing AbSplice (v1.0.0).

**Comparison of juncmut with SAVNet.** We conducted a comparison between juncmut and SAVNet. First, we performed SAVNet on the 445 pair genome and transcriptome sequencing data. The VCF files were downloaded as the previous subsection and restricted to those with ≤ 0.01 allele frequencies. We processed transcriptome sequencing data as performed in the previous study to generate junction and intron-retention files[3].

To ensure a fair comparison between juncmut and SAVNet, we excluded variants identified by SAVNet if (1) the corresponding splicing junction is included in the control panel constructed from the GTEx transcriptome, (2) the support read count for the splicing junction is two or fewer, or (3) the proportion of the splicing junction is less than 0.05. Additionally, we restricted to those splicing-associated variants in SAVNet that exhibit a pattern of substitution within novel splice motifs, as targeted by juncmut. Finally, we assessed the overlap of the variants identified by SAVNet and juncmut.

## Evaluation of juncmut using TCGA whole-genome and transcriptome

**Classification of splice-site creating variants using tumor and matched-control whole-genome sequencing data.** We limited our comparison to the TCGA transcriptome sequence data that have corresponding high-coverage whole-genome sequencing data from the PCAWG project (Supplementary Data 2). We downloaded the BAM files from https://registry.opendata.aws/icgc/ and aligned them to the reference genome. We then measured sequencing depths, the numbers and the ratios of SSCV supporting reads. Subsequently, we classified them into "Germline," "Somatic," "Somatic or germline," "Ambiguous," and "False positive" categories according to the flowchart presented in Supplementary fig. 6.

**Comparison of juncmut with SAVNet.** We conducted a comparison between juncmut and SAVNet, restricting our analysis to TCGA transcriptome data employed in the PCAWG Transcriptome project[25]. The results from SAVNet were derived from previous studies, and converted from GRCh37 to GRCh38 genome coordinates with the liftOver tool (https://genome-store.ucsc.edu/).

To ensure a fair comparison between juncmut and SAVNet, we excluded splicing-associated variants detected by SAVNet, as was done in the comparison using 1000 Genomes Project data. For juncmut, we confined our analysis to SSCVs classified as "somatic," aligning with SAVNet, which exclusively targets somatic variants. Finally, we assessed the overlap of the variants identified by SAVNet and juncmut.

## Investigation of positional relationship between splice-site creating variants and Alu sequences

First, we obtain the RepeatMasker file from the UCSC Genome Browser (https://hgdownload.soe.ucsc.edu/goldenPath/hg38/database/rmsk.txt.gz) and extract records where the "repFamily" field is set to "Alu." We then create a BED file using the genomic position information ("genoName," "genoStart," and "genoEnd"). Next, we generate a BED file for the SSCVs. Using bedtools[57] version 2.31.0 with the command 'bedtools intersect,' we identify SSCVs that intersect with Alu sequences as well as their genomic positions and subclass information.

Next, for each Alu sequence intersecting with SSCVs, we extract the DNA sequence and perform pairwise alignment to a reference Alu sequence constructed in a previous literature[39] using the Align.PairwiseAligner.align function of Biopython version 1.81[58]. Utilizing this alignment information, we convert the coordinates of the SSCV position, the primary novel SS, and the secondary SS (for SSCVs resulting in cryptic exons) in the human reference genome to the coordinate system in the reference Alu sequence.

## Compilation of disease-associated genes

We downloaded the ClinVar VCF file from the ClinVar FTP site (https://ftp.ncbi.nlm.nih.gov/pub/clinvar/vcf_GRCh38/) as of December 11, 2022. We extracted pathogenic variants, which are defined as those whose CLNSIG INFO key is either of "Pathogenic," "Likely_pathogenic," or "Pathogenic/Likely_pathogenic" in this study. Here, we also imposed additional constraints, CLNSTAT INFO key is either of "criteria_provided,_multiple_submitters,_no_conflicts," "reviewed_by_expert_panel," or "practice_guideline" to focus on variants with solid evidence. We then compiled a list of genes with at least one pathogenic variant. We integrated this gene list with those included in those listed in the Cancer Gene Census version 97 (https://cancer.sanger.ac.uk/census), and ACMG SF v3.2 list[42].

## Cell lines

PC-9 cell line was obtained from Dr. Nishio Kazuto (Kindai University, Osaka, Japan) in 2005 and authenticated in 2022 using the Promega GenePrint 10 System (BEX). Leukemia Jurkat cell line was purchased

from RIKEN BioResource Research Center in 2022. PC-9 and Jurkat cells were grown in RPMI1640 (Gibco) with 10% FBS (Gibco) and 1% penicillin/streptomycin (Wako). Lenti-X 293 T cell lines were purchased from Takara in 2021 and were cultured in DMEM (Gibco) with 10% FBS and 1% penicillin/streptomycin. All cell lines were tested negative for mycoplasma using Mycoplasma Plus PCR Primer Set (Agilent).

## Lentiviral transfection and mini-gene assay
*PIK3R1* (exon 11, intron 11, and exon 12) and *BCOR* (exon 10, intron 10, and exon 11) mini-gene constructs were generated in pLV lentiviral vectors (VectorBuilder). The constructs were cotransfected with LVpro Packaging Mix vectors (Takara) into Lenti-X 293 T cell lines using TransIT-293 Reagent (Takara) to produce viral particles. PC-9 cells were infected using RetroNectin (Takara) and were selected in 1 μg/ml puromycin (Nacalai) for three days. RNA was extracted from PC-9 cells stably expressing *PIK3R1* or *BCOR* mini-gene using RNeasy Plus Mini Kit (Qiagen) and cDNA was synthesized using ReverTra Ace qPCR RT Master Mix with gDNA Remover (TOYOBO). To amplify only mini-gene-derived but not endogenous transcripts of *PIK3R1* and *BCOR*, primers were designed to include the WPRE (woodchuck hepatitis virus posttranscriptional regulatory element) region which is specific to the vector sequence. Primers are listed in Supplementary Data 6.

## Genome editing using CRISPR-Cas9
*NOTCH1* SSCVs (c.5048-132 G > C, c.5048-132 G > T) were created in the PC-9 cell lines as previously reported[49,50]. We designed sgRNA and donor templates for CRISPR-Cas9 homology directed repair (HDR) using Alt-R HDR Design Tool (Integrated DNA Technologies, IDT). Briefly, we formed the ribonucleoprotein complex with 120 pmol Cas9 nuclease (IDT) and 150 pmol sgRNA (IDT). The reaction mixtures and 120 pmol donor templates were nucleofected into PC-9 cells ($1 \times 10^5$ cells) suspended in 20 μl of SE solution (Lonza) using 4D-Nucleofector (Lonza) with EN-138 mode. DNA was extracted from single clones using QuickExtract DNA Extraction Solution (Lucigen) and mutations were confirmed by Sanger sequencing (Azenta). Sequences of sgRNA, donor templates, and primers are listed in Supplementary Data 7.

## Antibodies and Western blot analysis
Cells were lysed with RIPA buffer (Nacalai) supplemented with cOmplete Mini EDTA-free Protease inhibitor cocktail (Roche) and PhoSTOP phosphatase inhibitor cocktail (Roche). The total cell lysate (20 μg) was subjected to 4-12% Bolt Bis-Tris Plus Mini Protein Gels (Invitrogen) and transferred to iBlot 2 PVDF membranes (Invitrogen). Membranes were scanned by ImageQuant LAS 4000 (GE Healthcare). Antibodies are listed in Supplementary Data 8.

## Designing antisense oligos and treatment
Mutant-selective antisense oligonucleotides (ASOs) with full phosphorothioate (PS) + 2'-O-Methoxyethyl (2'MOE) modifications (IDT) were designed as previously described. Control antisense oligo was also designed to be non-targeting in the human genome. PC-9 cells harboring *NOTCH1* SSCVs were treated with 0.1 μM ASOs using 0.45% FuGENE 4 K Transfection Reagent (Promega) for two days. ASO sequences are listed in Supplementary Data 9.

## Reporting summary
Further information on research design is available in the Nature Portfolio Reporting Summary linked to this article.

## Data availability
The list of splice-site creating variants are accessible through SSCV DB (https://sscvdb.io) and Zenodo (https://doi.org/10.5281/zenodo.14053979). Source data are provided with this paper.

## Code availability
The workflow of juncmut is available at GitHub (https://github.com/ncc-gap/juncmut). The code of the version used in this study is available at Zenodo (https://doi.org/10.5281/zenodo.14011414).

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

## Acknowledgements

This work is supported by Grant-in-Aid for Scientific Research [KAKENHI 21H03549 & 23K27690] and Grand-in-Aid from the Japan Agency for Medical Research and Development [Practical Research for Innovative Cancer Control: JP23ck0106790 & JP21ck0106641; iD3 Booster: JP22nk0101636], and JST FOREST Program JPMJFR2260. We used the super-computing resource provided by ROIS National Institute of Genetics. The results shown here are partly based upon data generated by TCGA Research Network (https://cancergenome.nih.gov/) and the Genotype-Tissue Expression (GTEx) Project. The authors thank Erika Kawasaki and Rika Murakami (Division of Molecular Pathology, National Cancer Center Research Institute) for technical assistance.

## Author contributions

Y.S. designed the study. N.I. and Y.S. developed the software for detecting splice-site creating variants. A.O. developed a platform for analyzing massive transcriptome sequence data deposited in the Sequence Read Archive, and operated this platform. A.O., with assistance from Y.S., developed the portal website for SSCVs. Y.S., N.I., and A.O. organized and interpreted the list of SSCVs. Y.K., with assistance from Y.Y., conducted all the wet-lab experiments, including the development of model systems using CRISPR editing, mini-gene assays, and splice-switching antisense oligonucleotide administration. K.C. provided computational assistance across various aspects of the project. Y.S., N.I., A.O., and Y.K. generated figures. Y.S. wrote the manuscript.

## Competing interests

The authors declare no competing interests.
