## [Transparent Peer Review file · Nature Communications]

Systematically developing a registry of splice-site creating variants utilizing massive publicly available transcriptome sequence data

Corresponding Author: Dr Yuichi Shiraishi

Version 0:

Reviewer comments:

Reviewer #1

(Remarks to the Author)

In this manuscript, the authors present a computational pipeline for identifying splice-site creating variants (SSCVs) using only transcriptome sequencing data. By applying the pipeline to 322,072 publicly transcriptome data, 30,130 SSCVs were identified. The study then focused on the classification and features of SSCVs, revealed the characteristics of Alu exonization via SSCVs, and identified the SSCVs in disease-related genes. Finally, the authors investigate SSCVs in the intronic region of the NOTCH1 gene and demonstrate the feasibility of suppressing their activation using splice-switching ASOs.

While the research and the establishment of the database (SSCV DB) could have certain contributions to the field, making it more convenient for others to study splicing-related SNPs. However, there are some major concerns. My first concern is the novelty of this work. The authors claimed that “variants that cause formations of novel splice-sites (splice-site creating variants, SSCVs) are particularly difficult to identify and often overlooked in genomic studies”, which I can not agree. In fact, among the variants that affect alternative splicing, the variants that generate new splice site are the easiest to identify through RNA-seq, because they produce new isoforms with novel splice junctions that can be easily mapped (like they did in this paper) and tested by reporter assay. In addition, the computational methods to score the splice site is pretty robust, which can be combined with the new splice junction from RNA-seq to accurately predict the so called “splice-site creating variants”. In addition, I also concern about the accuracy and broad application of their method since the RNA-seq data may contain many sequencing errors that may compromise their prediction. Further evidence is needed to solidify the feasibility of this approach.

Specific comments:

1. Related to the result “Most of the SSCVs (152 out of 153) were also detected as genomic variants by an investigation of raw whole-genome sequencing data, confirming that these SSCVs are indeed actual genomic-level variants”. It seems that the genome analysis used here has over-estimated the number of potential new splice site, and thus almost all SSCVs can be recovered from genome analysis. In another word, it is also good to know how many SSCVs that identified using the combined whole-genome sequencing analysis can also be identified using only the transcriptome.
2. Related to the result “juncmut detected about 23.1% (25 out of 108) of the SSCVs identified by SAVNet”, did the authors examine possible reasons why the remaining 77 percent were not detected? Does this result indicate that it is hard to have a comprehensive identification of SSCV using only transcriptome data?
3. The geomAD, which is used by most SNP researchers, has listed a lot of SNPs in the splice region. A comparative analysis between SSCV DB and geomAD would elucidate the advantages of the former.
4. How is the splice site strength of novel splice sites identified by “juncmut” compared with the annotated splice sites as judged by MaxEntropy scores or the PSI values? In addition, for the SNPs that creat novel splicing sites, are their allele frequency correlated with the strength of the novel splice site.
5. Regarding Figure 6, it would be useful to investigate whether splice-switching ASOs targeting NOTCH1 SSCVs exhibit any functional effects on cancer cells. Exploring the potential for intervention in cancer through SSCV interference would be of significant interest.
6. In addition, for the figure 6e, they should give an explanation on why two of the ASOs (ASO1, and ASO2) can inhibit the use of cryptic site (5048-132G>C), whereas the third one (ASO3) have a much weaker effect despite they probably bind to the mutated sites equally well. The same results were also observed for the G>T mutation. One possibility is that the AG rich element (GAAGAAGAG) at the downstream may function as splicing enhancer or silencer by binding to SR proteins, and

thus blocking this element alone (rather than block the splice site) will reverse the splicing mutation. The authors should examine this rather than ignoring additional splicing regulatory elements.

(Remarks on code availability)

Reviewer #2

(Remarks to the Author)

This study by Iida and co-workers presents a method and an associated resource to identify splice-site creating variants (SSCVs) solely from RNA-seq data. I found the method to be original, creative and well designed. Also, the authors prove its high specificity and decent sensitivity in a thorough and reliable way.

While the method itself may not be very commonly used in clinical settings, where it is more common to sequence the patient's genome (or exome), I believe that the resource the authors have built by applying their method to all RNA-seq datasets in SRA will be very useful to the genetics community. In addition, the downstream analyses they have done, specifically on Alu elements, disease-causing variants and the variants identified in NOTCH are also interesting and well performed.

Therefore, I found the study to be worth publishing. I have only a few comments/suggestions:

- The authors show that the introduction of the cryptic sequence in NOTCH enhances activation (Figure 6d). However, they did not show if the treatment with ASOs reduces this over-activation.
- It would be interesting to see which specific genetic diseases are enriched among the disease genes with detected SSCVs.
- The authors have used SpliceAI for a few comparisons. I was wondering if SpliceAI or similar tools could be added as another layer to their method to reduce false positives (estimated to be around 3-4%) or even increase sensitivity, by relaxing other cut-offs. I do not think this is a must, it is only a suggestion.
- "Transcriptional consequences": I found this misleading, as the consequences are not on transcription, but on the mature transcript structure and processing. Please re-word.

(Remarks on code availability)

Reviewer #3

(Remarks to the Author)

(Remarks on code availability)

Reviewer #4

(Remarks to the Author)

This study aimed to develop a novel framework to screen for splice site-creating variants (SSCVs) solely using transcriptome data available from the SRA. Although the findings presented are of potential interest, I believe this framework has a significant limitation: it relies on the sequenced reads generated from, in authors' words, "the heterogeneity of splicing effects and an imperfect penetrance of SSCVs (probably due to competition with existing splice sites), mismatch bases showing SSCVs appear to be frequently observed in the transcriptome (Page 4)", which might lead to a lower detection rate (only 23.1% of SSCV identified by SAVNet were detected) and preclude this registry from being a comprehensive annotation.

I have several issues that could be addressed by the authors.

Major points;

1) As this study involves secondary use of data obtained from previous research, it is important to compare the findings with those in many other preceding papers (e.g. Ferreira et al. Sci Rep 2016) to emphasize the progress brought by the present study. In addition, the authors mentioned only SpliceAI (and their own SAVNet) as pre-existing tools. Other tools, such as FRASER (Martes et al., Nat Commun 2021) and AbSlice (Wagner et al., Nat Genet 2023), which also evaluated the GTEx dataset, should be referenced, and their performance in detecting SSCVs could be compared.

2) Because the present framework utilizes transcriptome data, it is essential to examine the possible effects of RNA-editing on splicing. The genetic variants detected in the RNA-seq reads could be attributed to RNA-editing.

3) The present study also makes a claim about Alu exonization via SSCVs. However, this observation has already been reported using GTEx samples (Florea et al. Front Mol Biosci. 2021). Please reference this paper and consider toning down the claim in the abstract and main text.

4) Finally, the authors showed an intronic variant of the NOTCH1 gene as an example of novel disease-related SSCVs. However, the authors only demonstrated that this variant was detected in multiple cancer samples; I don't think this observation is enough to conclude that it is a disease-causing variant. If this variant is a germline variant, a significant difference in its frequency from the general population could be demonstrated. If it is somatic, its absence in the germline genome could be demonstrated.

Minor points

5) I highly recommend that the authors have the manuscript checked by a native English speaker. There are some grammatical errors that prevent this reviewer from precisely understanding the manuscript.

6) "1. The splicing junction produced by SSCVs is usually rare and not typically found in the general population.(Page 4)"
>>>

It is unclear whether the rarity lies in the splice junction itself in the RNA-seq reads or in the frequency of SSCVs in the general population (or both).

(Remarks on code availability)

Version 1:

Reviewer comments:

Reviewer #1

(Remarks to the Author)

In the revised version of this manuscript, the authors conducted additional analysis to address most of my concerns. They also made good efforts to give additional discussions to explain potential problems even when they can not provide a straightforward answer. I appreciate their effort, and feel that they should be given the benefit of the doubt for most of my technical concerns. After consult with my co-reviewer, we support the publication of this manuscript.

(Remarks on code availability)

Reviewer #2

(Remarks to the Author)

The authors have properly addressed my comments.

(Remarks on code availability)

Reviewer #3

(Remarks to the Author)

(Remarks on code availability)

Reviewer #4

(Remarks to the Author)

Thank you for responding to my previous comments. The authors' responses have substantially improved the manuscript. However, my initial assessment that the lower detection rate may prevent this registry from being a comprehensive annotation has not changed.

(Remarks on code availability)

Response to Reviewers' Comments

Reviewer 1

In this manuscript, the authors present a computational pipeline for identifying splice-site creating variants (SSCVs) using only transcriptome sequencing data. By applying the pipeline to 322,072 publicly transcriptome data, 30,130 SSCVs were identified. The study then focused on the classification and features of SSCVs, revealed the characteristics of Alu exonization via SSCVs, and identified the SSCVs in disease-related genes. Finally, the authors investigate SSCVs in the intronic region of the NOTCH1 gene and demonstrate the feasibility of suppressing their activation using splice-switching ASOs.

While the research and the establishment of the database (SSCV DB) could have certain contributions to the field, making it more convenient for others to study splicing-related SNPs.

Thank you very much for providing a great summary of our manuscript. I would like to address your concerns, in particular regarding the validity and novelty of our approach for detecting splice site creating variants in the following discussion.

However, there are some major concerns. My first concern is the novelty of this work. The authors claimed that “variants that cause formations of novel splice-sites (splice-site creating variants, SSCVs) are particularly difficult to identify and often overlooked in genomic studies”, which I can not agree. In fact, among the variants that affect alternative splicing, the variants that generate new splice site are the easiest to identify through RNA-seq, because they produce new isoforms with novel splice junctions that can be easily mapped (like they did in this paper) and tested by reporter assay.

First, many of the splicing-associated variants registered in ClinVar are of the type that disrupt existing splice sites, with very few variants that create novel splice sites. In our study as well, out of 5,121 SSCVs in disease-related genes, only 82 were registered in ClinVar, as demonstrated in the section of “Splice-site creating variants affecting disease-related genes” (on line 32 of page 7). This indicates that efforts to accurately predict and validate splice-site creating variants have not yet significantly advanced.

Certainly, we can relatively easily identify potential new splice sites by compiling a list of unannotated and infrequently occurring splice junctions. It should be noted, however, that these are not always created by genomic variants; they can also result from epigenetic changes or the regulation of splicing factors. Furthermore, as the reviewer suggested, apparently novel splice junctions may be sequencing errors or alignment artifacts.

To confirm that a novel splice junction is created by a genomic variant, further validation is necessary. A reporter assay may be a great choice for validation if the number of possible novel splice sites is limited. **Nevertheless, conducting reporter assays for the tens of**

thousands of SSCV candidates extracted from 322,072 publicly available transcriptome datasets would be a formidable task.

Our approach can extract SSCVs purely computationally in a thorough and careful manner, making the most of the massive publicly available transcriptome sequencing data for screening.

In addition, the computational methods to score the splice site is pretty robust, which can be combined with the new splice junction from RNA-seq to accurately predict the so called “splice-site creating variants”.

There are various methods for scoring splice sites, such as MaxEnt Score. The combination of splicing junctions obtained from RNA-seq with splice site scoring methods and given genome data results in a vast number of possibilities. The following approaches would naturally be considered:

1. Obtain genomic mutation profiles from genome sequence data using tools such as GATK (for germline mutations) or Mutect2 or Strelka2 (for somatic mutations).
2. Identify unannotated edges of splice junctions from RNA sequence data, which can be considered as potential novel splice sites, and obtain a list of these novel splice sites.
3. Compile the pair of novel splice sites and closely located genomic mutations. Then, evaluate the difference in scores (e.g., MaxEntScore) for novel splice site candidates before and after the associated genomic mutation, and extract the mutations that increase the score beyond a certain threshold.

In addition, our previous approach, SAVNet (Shiraishi et al., Genome Research, 2018), assesses whether the supporting read counts of splice junctions are significantly higher in individuals with the candidate splicing-causing genomic mutation. Other approaches also employ the combinations of splice junctions, sample specificity, and splice site scoring, although the implementation details differ.

However, previous approaches have assumed that both genome and transcriptome sequence data are available. To the best of our knowledge, there has been no other approach to identify splice-site creating variants using only transcriptome sequence data. We carefully designed the algorithm and evaluated our approach using public data such as 1000 Genomes, GTEx, and TCGA. Additionally, we performed SSCV screening by applying our methods to over 300,000 publicly available transcriptome sequencing datasets. In that sense, we argue that our method has a solid novelty.

In addition, I also concern about the accuracy and broad application of their method since the RNA-seq data may contain many sequencing errors that may compromise their prediction. Further evidence is needed to solidify the feasibility of this approach.

Certainly, RNA sequencing is prone to sequence errors. Furthermore, alignment artifacts present a significant challenge, especially around novel splice sites, because alignment tools such as STAR use known isoform information. Nonetheless, in the junctmut approach, by implementing several filtering steps—restricting to genomic variants that explain splice junctions and eliminating alignment artifacts through realignment—we believe many of these sequence

errors could be mitigated in the identification of SSCVs. Indeed, through evaluations using datasets from the 1000 Genomes Project, which provide paired genome and transcriptome data, we confirmed that most SSCVs identified in the transcriptome via junctmut are consistently validated at the genomic level.

Specific comments:

1. Related to the result “Most of the SSCVs (152 out of 153) were also detected as genomic variants by an investigation of raw whole-genome sequencing data, confirming that these SSCVs are indeed actual genomic-level variants”. It seems that the genome analysis used here has over-estimated the number of potential new splice site, and thus almost all SSCVs can be recovered from genome analysis. In another word, it is also good to know how many SSCVs that identified using the combined whole-genome sequencing analysis can also be identified using only the transcriptome.

Here, we examined precision, specifically how many of the SSCV candidates detected by junctmut are truly creating novel splice sites and causing aberrant splicing alterations. To address this, we conducted a two-step validation. First, we confirmed that the SSCV candidates detected by junctmut from RNA-seq data are indeed genomic mutations, rather than sequencing errors, alignment artifacts, or RNA editing (152 out of 153 SSCV candidates were validated). Second, we demonstrated that these mutations actually cause splicing abnormalities using data from GTEx (23 out of 24 candidates in which at least one individual in GTEx had genomic variants in the whole-genome sequences were validated).

Regarding the comparison with “SSCVs identified using the combined whole-genome sequencing analysis,” we used splice-site creating variants detected by SAVNet from GEUVADIS whole genome and RNA-seq as the reference data, as we did in the following TCGA section. Among the 165 SSCVs identified by SAVNet, 33.3% SSCVs (55) could be detected by junctmut, which was slightly higher compared to those in TCGA. One potential explanation for this may be that the genomic variants in GEUVADIS are germline, and thus exhibit higher variant allele frequencies and levels of smaller splicing changes than somatic variants, which are primary the focus of TCGA data. We believe these factors lead to the higher sensitivity of junctmut in this dataset. We included the evaluation on line 18 of page 4 in the manuscript as follows:

In addition, we performed a comparison with our previous approach, SAVNet³, which collects splicing-associated variants using paired genome and transcriptome data. Despite using only transcriptome data, junctmut detected approximately 33.3% (55 out of 165) of the SSCVs (restricted to those with ≤ 0.01 allele frequency) identified by SAVNet (Supplementary Figure 5a).

We also included the Venn diagram in Supplementary Figure 5a (also included in the later pages of this rebuttal). We also modified the description on line 30 of page 4 as follows:

We also performed a comparison with SAVNet for this dataset, where junctmut was able to detect approximately 23.1% (25 out of 108) of the somatic SSCVs identified by SAVNet (Supplementary Figure 5b)²⁵. The slightly lower sensitivity compared to the 1000 Genomes Project dataset may be due to the fact that our reference data is limited to somatic variants,

which often have lower variant allele frequencies and smaller splicing changes than germline variants.

We furthermore included the SAVNet comparison procedure on line 7 of page 15 in the manuscript.

2. Related to the result “juncmut detected about 23.1% (25 out of 108) of the SSCVs identified by SAVNet”, did the authors examine possible reasons why the remaining 77 percent were not detected? Does this result indicate that it is hard to have a comprehensive identification of SSCV using only transcriptome data?

Thank you very much for your valuable suggestion. We have investigated the reasons why 83 SSCVs were identified by SAVNet using both genome and transcriptome sequencing data. First, the juncmut procedure requires that there are two or more mismatch bases accounting for the primary novel splicing junction, and their variant allele frequency (VAF) must be at least 5% when performing a pileup at that location. However, 66 variants did not meet this criterion. Additionally, nine variants were removed at various stages, such as due to an excess of intersecting splicing junctions with the primary novel SJ, or during the realignment of short-read validation. Three variants were not identified due to the reference sequence changes resulting from coordinate conversion from GRCh37 to GRCh38 via liftover. Two variants did not exhibit the substitution pattern targeted by juncmut (see the "Identification of variants that account for the corresponding aberrant splicing junctions" subsection on line 26 of page 11). The remaining three variants were excluded due to inconsistencies in transcriptome annotation.

From the stage of previous submission, we mentioned in the Discussion section as:

A limitation of our approach is inherent biases, and low sensitivity compared to the approach using both genome and transcriptome. This is because juncmut relies on a limited number of short reads with mismatches corresponding to SSCVs, arising from the heterogeneity of splicing abnormalities in transcriptome sequence. When exploring splicing-associated variants in individual patients for clinical purposes, we recommend a combination of juncmut and other approaches involving both genome and transcriptome sequence data^{3,23,50}. However, the ability to acquire a catalog of SSCVs through reanalysis of existing transcriptome sequence data is an attractive feature. Particularly because juncmut can be performed on individual transcriptome sequence data, execution on large-scale transcriptome sequences is extremely convenient.

As mentioned, it is challenging for juncmut to detect a "comprehensive" list of SSCVs, and we sincerely reported our sensitivity evaluation using TCGA data as 22.7%. However, the advantage of the juncmut approach is that it can extract SSCVs solely from RNA-seq data. This capability allowed us to utilize over 300,000 RNA-seq samples (which are still increasing rapidly) and detect as many as 30,000 SSCVs purely through computational procedures. This catalog included 5,121 SSCVs affecting disease-associated genes. Also, we identified novel gain-of-function SSCVs in the *NOTCH1* gene, which could be suppressed by splicing-switching ASOs. Even with somewhat lower sensitivity, by maximizing utility, we can effectively construct an enormous catalog of SSCVs.

3. The geomAD, which is used by most SNP researchers, has listed a lot of SNPs in the splice region. A comparative analysis between SSCV DB and geomAD would elucidate the advantages of the former.

Thank you for your important feedback. We divided the allele frequencies in gnomAD into bins with breakpoints at $1e-5$, $1e-4$, $1e-3$, and $1e-2$, and counted the number of detected SSCVs within each gnomAD allele frequency bin. Please find Supplementary Figure 10 in the new submission (also included on the later pages of this rebuttal). From this analysis, it became apparent that the number of variants in gnomAD increases as the allele frequency decreases. In contrast, the SSCVs showed a certain degree of bimodality, with a substantial number of SSCVs found even in regions with relatively high allele frequencies (e.g., $1e-3$ to $1e-2$). Additionally, many variants not identified in gnomAD (frequency = 0) were also detected. These observations suggest two key points.

First, SSCVs with relatively higher allele frequencies (in the range of $1e-3$ to $1e-2$) were detected more frequently, likely because transcripts containing these SSCVs are more likely to be analyzed, making them easier to detect. On the other hand, the Sequence Read Archive (SRA) transcriptomes include many cancer samples, leading to the detection of a considerable number of somatic SSCVs. The fact that many variants were not detected in gnomAD may be attributed to the presence of these somatic SSCVs. We have also included the following description on line 13 of page 5 in the manuscript:

Additionally, 18,357 (60.9%) were not registered in the gnomAD database (Supplementary Figure 10).

4. How is the splice site strength of novel splice sites identified by “juncmut” compared with the annotated splice sites as judged by MaxEntropy scores or the PSI values? In addition, for the SNPs that create novel splicing sites, are their allele frequency correlated with the strength of the novel splice site.

We calculated the MaxEnt score for the novel splice sites created by the SSCVs and their corresponding annotated splice sites (referred to as hijacked SS in this paper) and created the new Supplementary Figure 11 (also included in the later pages of this rebuttal). The novel splice sites created by SSCVs (Novel SS with SSCV in the figure) tend to have higher MaxEnt scores compared to those before the variant (Novel SS without SSCV), making them comparable to the annotated splice sites (Hijacked SS in the figure). We have also included the following description on line 14 of page 5 in the manuscript:

As expected, the novel splice sites created by SSCVs increased MaxEnt scores, making them comparable to the hijacked SSs (originally utilized splice-sites hijacked by the SSCVs, see Supplementary Figure 11).

We also investigated the relationships between allele frequencies (gnomAD) and MaxEnt scores. As shown in Reviewer Only Figure 1 (included in the later pages of this rebuttal), there were essentially no relationships between the allele frequencies and MaxEnt scores.

5. Regarding Figure 6, it would be useful to investigate whether splice-switchi

ng ASOs targeting NOTCH1 SSCVs exhibit any functional effects on cancer cells. Exploring the potential for intervention in cancer through SSCV interference would be of significant interest.

We appreciate the constructive advice on evaluating any functional effects on cancer cells.

The cleavage of the NOTCH1 juxtamembrane expansion (JME) produces the NOTCH intracellular domain (NICD). NICD interacts with a transcription factor complex, resulting in subsequent activation of a broad range of target genes. Therefore, it is difficult to show a unique phenotype related to NOTCH signals.

Instead, we added a western blot analysis of NICD protein, a key indicator of NOTCH activation, in genome-edited models treated with ASOs. Treatment with ASOs successfully decreased the expression of NICD (new Figure 6f). Considering that this paper places a strong emphasis on using a computational approach to screen and interpret SSCVs, we believe that it is a valuable addition that one of the gain-of-function SSCVs, NOTCH1, was functionally validated by using CRISPR models and ASOs, both in terms of mRNA and protein. Further experiments on NOTCH1 would be beyond the scope of this paper.

6. In addition, for the figure 6e, they should give an explanation on why two of the ASOs (ASO1, and ASO2) can inhibit the use of cryptic site (5048-132G>C), whereas the third one (ASO3) have a much weaker effect despite they probably bind to the mutated sites equally well. The same results were also observed for the G>T mutation. One possibility is that the AG rich element (GAAGAAGAG) at the downstream may function as splicing enhancer or silencer by binding to SR proteins, and thus blocking this element alone (rather than block the splice site) will reverse the splicing mutation. The authors should examine this rather than ignoring additional splicing regulatory elements.

We previously examined ASO treatments in terms of exonic splicing enhancers (Kobayashi et al., Nature, 2022, PMID: 35236983). Therefore, we applied our previous method to predict ESE motifs around these regions in both the wild-type and *NOTCH1* c.5048-132G>C sequences. As this reviewer hypothesized, there are many enhancer motifs around the specific region covered by ASOs 1 and 2, but not 3. The relative strength of ESE and ESS also showed a tendency for enhancement (see Reviewer Only Figure 2 and 3).

We agree that this is a very interesting potential explanation for the different effects. However, in the field of designing ASOs, there is no unique strategy to precisely predict their effects. As a result, many researchers perform screenings to find the best ASOs. Given this situation, we need to be careful not to overstate the scientific significance of the simulated data.

The reviewer suggested that we try "blocking enhancer elements alone (rather than blocking the splice site)." However, we believe that the advantage of ASOs in the field of cancer is their ability to design mutant-selective ASOs that influence only the mutant allele and not the wild-type allele, resulting in potentially mild toxicity in a clinical setting as shown in our previous Nature paper. Therefore, it is important to design mutant-selective ASOs targeting *NOTCH1* c.5048_132G>C or G>T mutations rather than targeting only ESEs out of the mutations.

Supplementary Figure 5: Venn diagrams showing the overlap of SSCVs identified by juncmut and the previous approach (SAVNet). (a) Comparison using 1000 Genomes Project whole genome and transcriptome sequencing data. (b) Comparison using PCAWG whole genome and transcriptome sequencing data. Refer to the Methods section for details regarding the comparison method.

Supplementary Figure 10 (in new submission): The number of SSCVs and variants in gnomAD, categorized by allele frequency (AF) in gnomAD v3.0. We divided the allele frequencies in gnomAD into bins with breakpoints at 1e-5, 1e-4, 1e-3, and 1e-2, and counted the number of detected SSCVs within each gnomAD allele frequency bin. Please note that we did not focus on SSCVs with high allele frequencies (>0.01), which were excluded from this analysis.

Supplementary Figure 11 (in new submission): Distributions of MaxEnt scores for the novel SS created by SSCVs (Novel SS with SSCV) and their corresponding annotated SS (Hijacked SS). MaxEnt scores for the novel SS before the variants occur are also shown (Novel SS without SSCV).

Reviewer Only Figure 1: Distributions of MaxEnt scores for the novel SS created by SSCVs, stratified by the allele frequencies provided from gnomAD.

NOTCH1 reference (wild type)

Reviewer Only Figure 2: This figure shows the predicted Exonic Splicing Enhancer (ESE) and Exonic Splicing Silencer (ESS) sites, as well as the dominance between ESE and ESS, around the SSCVs of *NOTCH1* gain-of-function mutations (c.5048-132G>C, c.5048-132G>T) and the Antisense Oligonucleotides (ASOs) designed for these regions. The sequence shown in this figure represents the wild-type case, where no SSCV mutations are present.

NOTCH1 c.5048-132G>C

Reviewer Only Figure 3: This figure shows the predicted Exonic Splicing Enhancer (ESE) and Exonic Splicing Silencer (ESS) sites, as well as the dominance between ESE and ESS, around the SSCVs of *NOTCH1* gain-of-function mutations (c.5048-132G>C, c.5048-132G>T) and the Antisense Oligonucleotides (ASOs) designed for these regions. The sequence shown in this figure represents the mutant-case, where the SSCV (c.5048-132G>C) is induced into the reference genome.

Reviewer 2

This study by Iida and co-workers presents a method and an associated resource to identify splice-site creating variants (SSCVs) solely from RNA-seq data. I found the method to be original, creative and well designed. Also, the authors prove its high specificity and decent sensitivity in a thorough and reliable way.

While the method itself may not be very commonly used in clinical settings, where it is more common to sequence the patient's genome (or exome), I believe that the resource the authors have built by applying their method to all RNA-seq datasets in SRA will be very useful to the genetics community. In addition, the downstream analyses they have done, specifically on Alu elements, disease-causing variants and the variants identified in NOTCH are also interesting and well performed.

Therefore, I found the study to be worth publishing. I have only a few comments/suggestions:

I deeply appreciate your review of my manuscript and the positive evaluation.

1. The authors show that the introduction of the cryptic sequence in NOTCH enhances activation (Figure 6d). However, they did not show if the treatment with ASOs reduces this over-activation.

Thank you very much for the constructive feedback. We have included a Western blot analysis of the NICD protein, a crucial marker of NOTCH activation, in our genome-edited models treated with ASOs. The treatment with ASOs effectively reduced NICD expression (new Figure 6f).

Figure 6 (f) Western blot analysis of the NOTCH intracellular domain (NICD) in CRISPR-edited clones treated with indicated ASOs for two days.

2. It would be interesting to see which specific genetic diseases are enriched among the disease genes with detected SSCVs.

This is what we have wanted to do for a long time, and we have been contemplating an effective method since the previous paper (Shiraishi et al., Nature Communications, 2022) to detect intron retention-associated variants from transcriptome data available in the Sequence Read Archive.

One of the biggest obstacles is that the metadata in the Sequence Read Archive is not well-organized, making it difficult to associate variants with phenotypes, such as specific diseases. Furthermore, the number of SSCVs is relatively small; even the most prominent case involves only 22 SSCVs in the ATM gene, which often does not yield statistically significant results.

At best, it is relatively straightforward to systematically extract the abstracts linked to each study in the sequence data. However, it is challenging to systematically extract information about each individual sample within those studies. For instance, even in studies related to cancer research, it is difficult to systematically determine whether a given sample is actually a cancer sample or a control. Therefore, as a preliminary experiment, we investigated which words were enriched in the abstracts of studies linked to samples with SSCVs. While there was one example, such as glioblastoma being associated with RB1 (Reviewer Only Figure 4), overall, the approach did not yield significant results.

We would like to aim to explore systematic annotation methods for the Sequence Read Archive, particularly focusing on individual sample annotations. This may involve leveraging large language models (LLMs) or other innovative approaches.

3. The authors have used SpliceAI for a few comparisons. I was wondering if SpliceAI or similar tools could be added as another layer to their method to reduce false positives (estimated to be around 3-4%) or even increase sensitivity, by relaxing other cut-offs. I do not think this is a must, it is only a suggestion.

Thank you for your important feedback. Our evaluation of the SSCVs detected in GEUVADIS using GTEx RNA-seq data revealed that, out of the 24 SSCVs we assessed, one was considered a false positive because no significant changes were observed in the RNA-seq data from GTEx. Notably, this SSCV had a SpliceAI score of 0.5. Additionally, three SSCVs with SpliceAI scores of less than 0.1 still exhibited significant changes in RNA (please see Supplementary Figure 3). This suggests that further validation is necessary to determine whether the use of SpliceAI consistently improves precision.

However, we firmly believe that for individual SSCVs, especially those of biological or medical interest, the SpliceAI score serves as a highly valuable metric. Therefore, we have decided to include SpliceAI scores for all SSCVs listed in our portal's SSCV database.

4. "Transcriptional consequences": I found this misleading, as the consequences are not on transcription, but on the mature transcript structure and processing. Please re-word.

Thank you very much for the important suggestion. We modified the term to "splicing consequences."

Reviewer Only Figure 4: This word cloud representing the words enriched in the descriptions listed in the projects corresponding to the samples in which SSCVs were detected in the RB1 gene.

Reviewer 3

This study aimed to develop a novel framework to screen for splice site-creating variants (SSCVs) solely using transcriptome data available from the SRA. Although the findings presented are of potential interest, I believe this framework has a significant limitation: it relies on the sequenced reads generated from, in authors' words, "the heterogeneity of splicing effects and an imperfect penetrance of SSCVs (probably due to competition with existing splice sites), mismatch bases showing SSCVs appear to be frequently observed in the transcriptome (Page 4)", which might lead to a lower detection rate (only 23.1% of SSCV identified by SAVNet were detected) and preclude this registry from being a comprehensive annotation.

Thank you very much for the insightful review.

In the Discussion section, we have already noted that our approach has certain limitations as follows in the previous submission:

A limitation of our approach is inherent biases, and low sensitivity compared to the approach using both genome and transcriptome. This is because junctmut relies on a limited number of short reads with mismatches corresponding to SSCVs, arising from the heterogeneity of splicing abnormalities in transcriptome sequence. When exploring splicing-associated variants in individual patients for clinical purposes, we recommend a combination of junctmut and other approaches involving both genome and transcriptome sequence data^{3,23,50}. However, the ability to acquire a catalog of SSCVs through reanalysis of existing transcriptome sequence data is an attractive feature. Particularly because junctmut can be performed on individual transcriptome sequence data, execution on large-scale transcriptome sequences is extremely convenient.

As stated, detecting a "comprehensive" list of SSCVs with junctmut is challenging, and we have transparently reported our sensitivity evaluation as 22.7%. Nevertheless, the key benefit of the junctmut approach is its ability to identify SSCVs exclusively from RNA-seq data via purely computational procedures. This allows us to leverage an increasing number of RNA-seq samples. Despite the lower sensitivity, by maximizing the utility of publicly available data, we can effectively build an enormous catalog of SSCVs.

I have several issues that could be addressed by the authors.

Major points;

1) As this study involves secondary use of data obtained from previous research, it is important to compare the findings with those in many other preceding papers (e.g. Ferreira et al. Sci Rep 2016) to emphasize the progress brought by the present study. In addition, the authors mentioned only SpliceAI (and their own SAVNet) as pre-existing tools. Other tools, such as FRASER (Martes et al., Nat Commun 2021) and AbSlice (Wagner et al., Nat Genet 2023), which also evaluated the GTEx dataset, should be referenced, and their performance in detecting SSCVs could be compared.

Basically, we used the GEUVADIS (1000 Genomes) and GTEx data primarily for the purpose of validating the sensitivity and accuracy of junctmut and, therefore, did not delve deeply into biological discoveries using these data. However, after reading Ferreira et al., Sci Rep 2016, we found that their analysis presented in their Supplementary Figure 4 (although not discussed in

the main document) aligns with our observation that the distance between the novel and original exon boundary tends to be a multiple of three. Therefore, we have added the reference and modified the manuscript as follows (on line 24 of page 5):

*A pronounced tendency was observed for the positional differences between primary novel SSs and hijacked SSs to be multiples of three, **as also implied in a previous study**²⁸.*

Please note that we already referenced FRASER in the previous submission (50th reference in the previous submission) as follows:

When exploring splicing-associated variants in individual patients for clinical purposes, we recommend a combination of junctmut and other approaches involving both genome and transcriptome sequence data^{3,23,50}.

Although we believe FRASER is a very useful tool, it primarily detects aberrant splicing from RNA-seq data rather than genomic variants. Therefore, after identifying aberrant splicing from RNA-seq, we also need to confirm the existence of genomic variants from genome sequencing data that generate the aberrant splicing. Thus, the purpose of FRASER differs from our approach, which performs SSCV detection solely from RNA-seq.

We have compared junctmut with AbSplice, more specifically, AbSplice-DNA, which incorporates SpliceMaps as we did in the comparison with SpliceAI in Figure 1E. The result is shown in the new **Supplementary Figure 4** (also included in the later pages of this rebuttal). In fact, there was no improvement compared to SpliceAI. AbSplice utilizes machine learning to fit scores using rare variants that generate splicing degree outliers as training data, along with SpliceAI, SpliceMaps, and other sources as features. The fact that it does not outperform SpliceAI suggests that the training data and features used in this process may not align with the splice site creating variant detection performed in the current study. In addition to the procedure for performing AbSplice in the Methods section (see subsection “Comparative evaluation of junctmut against SpliceAI and AbSplice” on line 25 of page 14), we have included the following description in the section of “Evaluation of junctmut approach” in the main text (on line 17 of page 4):

We also compared another approach, AbSplice²⁴, which showed a similar trend to SpliceAI (see Supplementary Figure 4).

2) Because the present framework utilizes transcriptome data, it is essential to examine the possible effects of RNA-editing on splicing. The genetic variants detected in the RNA-seq reads could be attributed to RNA-editing.

Thank you for pointing this out. We initially suspected the possibility that the variants detected by junctmut might be due to RNA editing. However, out of the 153 SSCVs detected from the GEUVADIS transcriptome, 152 were indeed mutations at the genomic level. The remaining one was a C>T mutation, which is not indicative of RNA editing by ADAR (which would be detected as an A>G substitution). Therefore, we believe that the false positives detected by junctmut are unlikely to be due to RNA editing.

3) The present study also makes a claim about Alu exonization via SSCVs. However, this observation has already been reported using GTEx samples (Florea et al. Front Mol Biosci. 2021). Please reference this paper and consider toning down the claim in the abstract and main text.

Thank you for informing me about the previous paper (Florea et al., Front Mol Biosci. 2021). I have added this paper to our citations.

Alu exonization is a well-known mechanism, and there are various themes and topics in its analysis. In Florea's paper, they identified transcripts containing Alu sequences based on a method of transcript assembly and discovered non-reference exonization resulting from the insertion of Alu sequences not present in the reference. I find their paper very interesting. However, this paper does not analyze the relationship between mutations (or SSCVs) and Alu exonization.

Our focus is specifically on analyzing where SSCVs create novel exons in the already inserted Alu sequences in genes (hotspots of novel splice donor/acceptor sites in the Alu sequences). This type of analysis is not conducted in Florea et al.'s paper, nor are we aware of any studies that examine new exon hotspots with the resolution of our paper. Therefore, we have determined to keep our current claims as they are.

4) Finally, the authors showed an intronic variant of the NOTCH1 gene as an example of novel disease-related SSCVs. However, the authors only demonstrated that this variant was detected in multiple cancer samples; I don't think this observation is enough to conclude that it is a disease-causing variant. If this variant is a germline variant, a significant difference in its frequency from the general population could be demonstrated. If it is somatic, its absence in the germline genome could be demonstrated.

We found two gain-of-function intronic SSCVs in the *NOTCH1* gene (chr9-136502620-C-G and chr9-136502620-C-A). For chr9-136502620-C-G, we identified it in two cases in TCGA transcriptome sequences: TCGA-A7-A13E (BRCA; breast cancer) and TCGA-NC-A5HR (LUSC; lung squamous carcinoma). These cases had matched whole genome sequencing data, allowing us to confirm that these are somatic variants, as shown in the new Supplementary Figure 23 (also included in the later pages of this rebuttal). For other transcriptome sequencing data without matched whole genome sequencing data, it is not possible to determine whether they are somatic. However, when we checked the population allele frequency using gnomAD 4.1.0, both genomic variants had an allele frequency of 0. Therefore, it is highly likely that these are somatic events in all the samples.

Minor points

5) I highly recommend that the authors have the manuscript checked by a native English speaker. There are some grammatical errors that prevent this reviewer from precisely understanding the manuscript.

I had my manuscript reviewed by Springer Nature's English Language Editing service, and corrected the overall grammatical errors.

6) "1. The splicing junction produced by SSCVs is usually rare and not typically found in the general population.(Page 4)"

>>>

It is unclear whether the rarity lies in the splice junction itself in the RNA-seq reads or in the frequency of SSCVs in the general population (or both).

Thank you very much. We have revised the description on line 9 of page 3 to make it easier to understand as follows:

SSCVs, particularly those associated with diseases, are rare in the population. Thus, the corresponding splicing junctions are typically not observed in the general population.

Supplementary Figure 4: Comparative evaluation of juncmut against AbSplice. We analyzed variants from the 1000 Genomes Project whole-genome sequencing data with AbSplice scores exceeding the thresholds (0.01, 0.05, and 0.2), as well as SSCVs identified by juncmut. We selected variants observed in at least one sample within the GTEx cohorts and calculated the combined p-value (measuring the difference in abnormal splicing ratios between samples with and without the variant across various tissues). Each p-value corresponding to these variants is plotted accordingly. The numbers of remaining SNVs with AbSplice scores exceeding 0.01, 0.05, and 0.2 were 14,301, 3,562, and 844, respectively; additionally, 22 SNVs identified by juncmut were plotted.

Supplementary Figure 23: Alignment views of the gain-of-function SSCV in *NOTCH1* (c.5048-132G>C). The upper panel shows the alignment for TCGA-A7-A13E (BRCA; breast cancer), while the lower panel shows the alignment for TCGA-NC-A5HR (LUSC; lung squamous carcinoma). Tumor and matched normal whole genome sequencing data were downloaded from the Genomic Data Commons.